# DEAD-box ATPase Dbp2 is the key enzyme in an mRNP assembly checkpoint at the 3'-end of genes and involved in the recycling of cleavage factors

Ebru Aydin [1], Silke Schreiner[1], Jacqueline Böhme[1], Birte Keil[1], Jan Weber [1], Bojan Žunar [2], Timo Glatter [3] & Cornelia Kilchert [1] ✉

mRNA biogenesis in the eukaryotic nucleus is a highly complex process. The numerous RNA processing steps are tightly coordinated to ensure that only fully processed transcripts are released from chromatin for export from the nucleus. Here, we present the hypothesis that fission yeast Dbp2, a ribonucleoprotein complex (RNP) remodelling ATPase of the DEAD-box family, is the key enzyme in an RNP assembly checkpoint at the 3'-end of genes. We show that Dbp2 interacts with the cleavage and polyadenylation complex (CPAC) and localises to cleavage bodies, which are enriched for 3'-end processing factors and proteins involved in nuclear RNA surveillance. Upon loss of Dbp2, 3'-processed, polyadenylated RNAs accumulate on chromatin and in cleavage bodies, and CPAC components are depleted from the soluble pool. Under these conditions, cells display an increased likelihood to skip polyadenylation sites and a delayed transcription termination, suggesting that levels of free CPAC components are insufficient to maintain normal levels of 3'-end processing. Our data support a model in which Dbp2 is the active component of an mRNP remodelling checkpoint that licenses RNA export and is coupled to CPAC release.

The expression of protein-coding genes depends on the production of functional mRNAs. In eukaryotic cells this involves the formation of the 3'-end of the mRNA by endonucleolytic cleavage and subsequent polyadenylation (CPA), a prerequisite for the export of mRNAs from the nucleus and their efficient translation in the cytoplasm[1–3].

CPA is carried out by the cleavage and polyadenylation complex (CPAC), consisting of cleavage and polyadenylation factor (CPF) and cleavage factors IA and IB (CFIA and CFIB) in yeast, and cleavage and polyadenylation specificity factor (CPSF), cleavage stimulatory factor (CstF) and the mammalian cleavage factors I and II (CFIm and CFIIm) in mammals. Despite differences in complex organisation and the

existence of divergent accessory factors, the core components and the general mechanism of CPA are highly conserved[1]. First, CPAC is recruited to an elongating RNA polymerase II (RNAPII)[4,5]. Components of CPAC then recognise consensus elements contained within the nascent transcript that define the polyadenylation site (PAS), including the canonical AAUAAA signal upstream of the cleavage site and various accessory elements that can be located either up- or downstream[6–10]. Recognition of the PAS triggers RNA cleavage by the CPAC-associated endonuclease (Ysh1 in yeast, CPSF-73 in humans). The polymerase module then adds a stretch of non-templated adenosines to the newly generated RNA 3'-end to form the poly(A) tail, a process that is

[1]Institute of Biochemistry, Justus-Liebig University Giessen, Giessen, Germany. [2]Department of Chemistry and Biochemistry, University of Zagreb Faculty of Food Technology and Biotechnology, Zagreb, Croatia. [3]Max Planck Institute for Terrestrial Microbiology, Marburg, Germany. ✉e-mail: cornelia.kilchert@chemie.bio.uni-giessen.de

controlled by poly(A)-binding proteins that associate with the nascent poly(A) tail[1,2,11]. Importantly, 3'-end processing is tightly coupled to transcription termination: The accessible 5'-PO$_4$ end of the downstream cleavage product is a substrate for the 5'-3' exonuclease Xrn2 (Dhp1 in *S. pombe*), which degrades the nascent RNA until it catches up with RNAPII, thereby helping to displace it from the DNA in a process termed torpedo-mediated transcription termination[12–15]. The coupling between 3'-end processing and transcription termination is further promoted by the activation of the CPAC-associated phosphatase Dis2/PNUTS-PP1 during PAS recognition. Dis2/PNUTS-PP1 then dephosphorylates the elongation factor Spt5 to slow down transcription in the termination zone, giving the torpedo nuclease an edge over RNAPII[16–18].

There is extensive crosstalk between RNA 3'-end formation and other steps of RNA processing, and the presence of various mRNA biogenesis factors can affect CPAC activity. For instance, splicing has a strong impact on 3'-end processing in mammals: Binding of U1 small nuclear ribonucleoprotein (snRNP) to RNA suppresses cleavage at upstream CPA sites[19,20], whereas interactions of U2 snRNP with CPAC stimulate cleavage activity[21–23]. At the same time, 3'-end processing is modulated by the nuclear RNA surveillance machinery, which can initiate hyperadenylation by the canonical poly(A) polymerase as a signal for RNA decay[24–28].

In contrast to the splicing reaction, in which intronic splice sequences are fully removed, CPA leaves most of the consensus elements that recruit CPAC intact. CPAC components have been found to crosslink to polyadenylated RNA in poly(A) interactome capture experiments in various organisms, including the fission yeast *Schizosaccharomyces pombe* (*S. pombe*)[29–34]. A role for poly(A)-binding proteins in preventing re-cleavage of polyadenylated transcripts has been proposed[35]. Evidence from yeast suggests that CPAC components remain with the processed transcripts until mRNA export factors are correctly incorporated into the ribonucleoprotein complex (mRNP) and trigger CPAC release, providing a checkpoint for mRNP assembly that links replacement of CPAC to export competence[36]. However, the factors that mediate CPAC release upon checkpoint completion have not been identified.

Potential candidates include DEAD-box ATPases, which have the ability to disrupt RNA-protein interactions. Dbp5 (DDX19 in humans), for example, releases export adaptors from the freshly exported mRNP upon its activation at the outer face of the nuclear pore complex, thus preventing re-entry of the mRNP into the nucleus[37–39]. DEAD-box ATPases belong to an abundant class of proteins that are involved in virtually all aspects of RNA metabolism and are found in all kingdoms of life[40–42]. There are at least 33 members of this protein family in humans; of the 19 annotated DEAD-box ATPases in *S. pombe*, most are essential for viability[43]. DEAD-box ATPases are conformationally flexible proteins: In the absence of ATP and RNA, they adopt an open state in which their two RecA-like domains, which are connected by a flexible hinge, can have a high degree of structural independence. Upon cooperative binding to ATP and single-stranded RNA, the RecA-like domains reorient to form the closed state of the active ATPase. ATP hydrolysis leads to RNA release[44,45]. When bound to a DEAD-box ATPase, the RNA substrate is forced into a kinked conformation that is incompatible with helical structures[46–49]. This mode of RNA binding is central to DEAD-box ATPase function: Distortion of the RNA can result in unwinding of short RNA duplexes (helicase activity) or destabilise RNA-protein interactions, allowing DEAD-box ATPases to release RNA-binding proteins and remodel mRNPs (RNPase activity)[50–54].

Despite a good understanding of the molecular function of DEAD-box ATPases, the high degree of coupling between different RNA processing pathways makes it challenging to link phenotypes that result from DEAD-box ATPase dysfunction – which are often pleiotropic – to a particular RNP remodelling event[55–57]. This also applies to Dbp2 (DDX5 in humans), which has been implicated in many different

processes, including transcription, pre-mRNA splicing, nuclear RNA export, RNA decay, RNA-DNA hybrid homoeostasis, regulation of liquid-liquid phase-separated compartments, gene looping, RNA silencing, and ribosome biogenesis[41,58,59]. Human DDX5, also known as p68, is involved in transcriptional regulation through its interaction with a histone acetyltransferase, p300/CBP, for which it is also a substrate[60–62]. Knockdown of DDX5 in HeLa cells results in the accumulation of unspliced pre-mRNA, and expression levels of DDX5 and its paralog DDX17 influence alternative splicing patterns[59,63–65]. This ability to regulate splicing extends to yeasts, where an (auto-)regulated splicing event in the *DBP2* gene controls Dbp2 expression[66,67]. In *S. cerevisiae*, Dbp2 preferentially crosslinks to RNAs that are targets for the Nrd1-Nab3-Sen1 (NNS) termination pathway[68,69], a budding yeast-specific alternative termination pathway for short stable non-coding RNAs[70,71]; at NNS target genes, Dbp2 was found to promote transcription termination[69]. In HeLa cells, read-through transcription was detected at the heat shock gene *HSP70* upon DDX5 depletion, with no observable reduction in PAS cleavage but retention of the *HSP70* transcript at the site of transcription[72]. In *Drosophila*, mutants of *Rm62*, the homologue of Dbp2, also fail to release *hsp70* RNA from the site of transcription and display a strong export defect reminiscent of mutations in the core mRNA export receptor *small bristles* (Mex67 in *S. pombe*, NXF1 in humans)[73].

Here, we report that Dbp2 is present at RNAPII-transcribed genes genome-wide in fission yeast, with a preference for the 3' end of genes. Using comparative proteomics, we find that Dbp2 interacts with CPAC and RNA export factors. We demonstrate that Dbp2 localises to cleavage bodies, subnuclear structures that are rich in CPAC components. Upon loss of Dbp2, 3'-processed, polyadenylated RNAs accumulate in the nucleus and in cleavage bodies, which is accompanied by a depletion of CPAC components from the soluble pool. Under these conditions, cells display an increased tendency to skip PAS sites and a delayed transcription termination, suggesting that the levels of free CPAC components are insufficient to maintain normal levels of 3'-end processing. Our data is consistent with a model in which Dbp2 is a key factor in an mRNP remodelling checkpoint that licenses RNA export and is coupled to CPAC release.

## Results
### Dbp2 is recruited to RNAPII-transcribed genes with a preference for the 3' ends
To characterise a potential function of Dbp2 in early RNA biogenesis in fission yeast, we carried out chromatin immunoprecipitation of endogenously HTP-tagged Dbp2 followed by sequencing (ChIP-seq). We observed efficient recruitment of Dbp2 to RNAPII-transcribed genes genome-wide, with a pattern that closely resembled that of an RNAPII-associated RNA processing factor, the SR-like protein Srp2, or of RNAPII itself (Fig. 1A). For both Dbp2 and Srp2, the extent of recruitment correlated with amounts of RNAPII at transcribed genes and RNA expression levels, suggesting that Dbp2 is recruited to transcribing RNAPII and/or nascent RNA (Fig. 1B and Suppl. Fig. 1A, B).

To determine the recruitment pattern across transcription units, we generated metagene plots comparing the distribution of Dbp2-HTP, Srp2-HTP, and RNAPII along ribosomal protein genes (RPGs) and protein-coding genes (Fig. 1C and Suppl. Fig. 1C, D). To be able to differentiate between stages of transcription (initiating, elongating, and terminating RNAPII), we compared to publicly available ChIP-seq data profiling different modifications of the C-terminal domain (CTD) of Rpb1, the largest subunit of RNAPII[74] (Fig. 1C and Suppl. Fig. 1C, lower panel), with serine 5 phosphorylation (S5P) and serine 2 phosphorylation (S2P) characteristic for initiating and terminating RNAPII, respectively[75,76]. The mean ChIP-seq coverage of both Dbp2 and Srp2 increases gradually along gene bodies. However, while the Srp2-HTP coverage peaks within the gene body and then decreases towards the

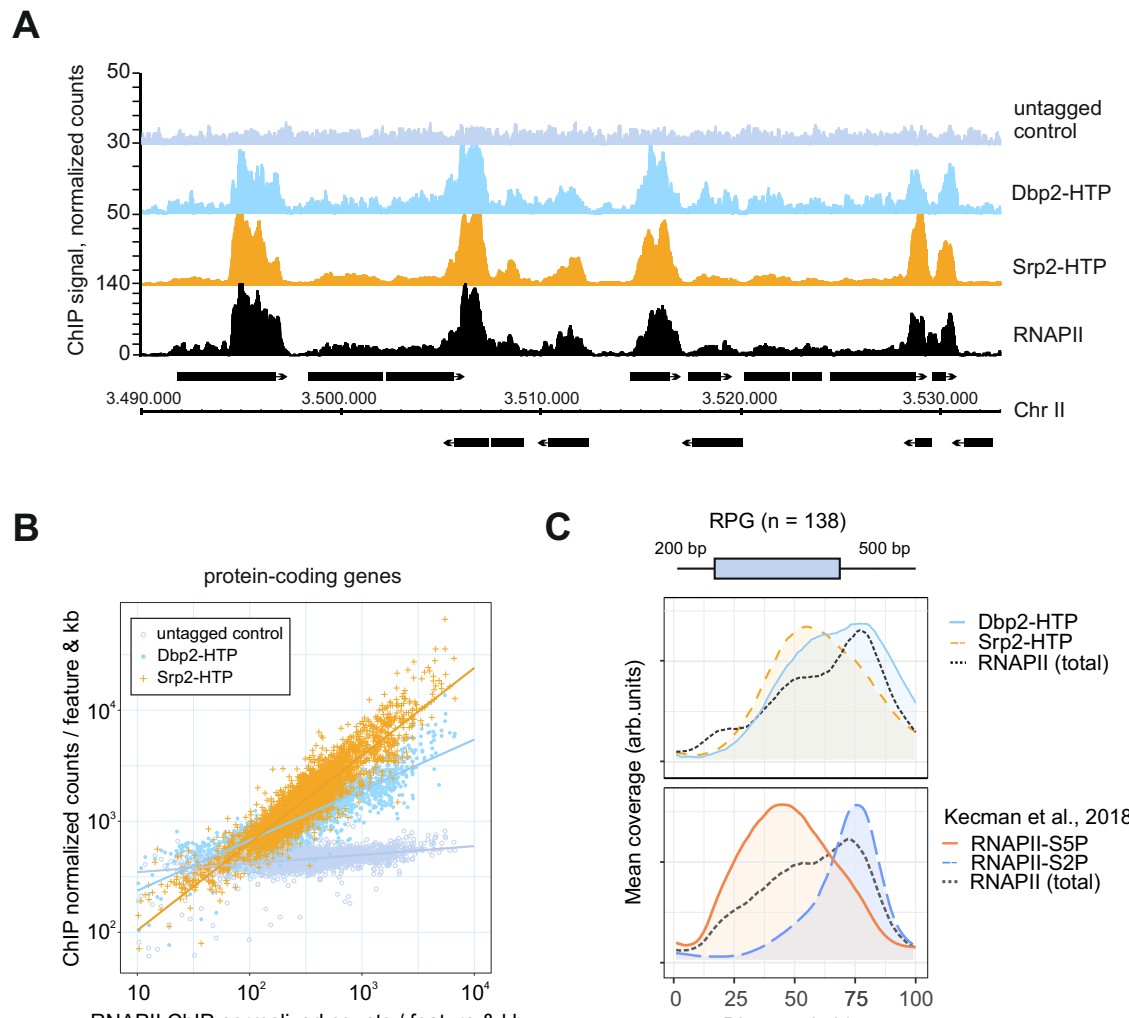

**Fig. 1 | Dbp2 is recruited to RNAPII-transcribed genes genome-wide.**
**A** Representative ChIP-seq traces of Dbp2-HTP and Srp2-HTP association with chromatin across a region of *S. pombe* chromosome II. An untagged wild type was included as control. RNAPII-ChIP signal for an isogenic wild type is shown for reference (α-Rpb1, 8WG16) (Kecman et al., 2018; GEO: GSE111326). Positions of protein-coding genes are indicated as black boxes. **B** Integrated counts of Dbp2-HTP and Srp2-HTP ChIP-seq signal across protein-coding genes relative to RNAPII, given as average counts per feature and kb * 1,000,000 (n = 3 biological replicates). Only genes with an RNAPII occupancy > 10 normalised counts per feature and kb per million were included. Trendlines were fitted using linear regression. RNAPII ChIP data from Kecman et al., 2018; GEO: GSE111326. Source data are provided as a Source Data file. **C** Metagene analysis of mean ChIP-seq coverage of Dbp2-HTP and

Srp2-HTP (n = 3 biological replicates), and total RNAPII (α-rpb1 (8WG16) in Dbp2-3myc; n = 2 independent experiments) (upper panel) across ribosomal protein genes (RPGs) including 200 bp upstream and 500 bp downstream of the annotated transcription units. Mean coverage given in arbitrary units; to compensate for differences in ChIP capability, mean coverage was adjusted by a constant scaling factor. Non-scaled curves with confidence intervals are provided in Suppl. Fig. 1D. Total Rpb1, Rpb1-S5P, and Rpb1-S2P ChIP for an isogenic wild type are included as reference (lower panel; data from Kecman et al., 2018; GEO: GSE111326). The schematic of the gene above the plot corresponds to an RPG of median length. Within the metagene, the positions of transcription start and end sites are distributed around the given coordinate because of the varying feature compression depending on gene length.

3′-end, the coverage for Dbp2-HTP reaches its highest point after the CPA site, coinciding with the RNAPII-S2P peak (Fig. 1C). Our data are compatible with a function for Dbp2 as a basal regulator of RNAPII transcription or co-transcriptional RNA processing, particularly at the 3′ end of genes.

### Comparative interaction profiling of Dbp2 and Srp2 reveals their association with RNAPII at different stages of the transcription cycle

We next sought to identify direct protein-protein interactors of Dbp2. Initial purification attempts without crosslinking did not result in the co-purification of significant amounts of proteins or protein complexes with Dbp2-HTP. This behaviour is reminiscent of other DEAD-box ATPases such as Dbp5, whose interactions with its regulators have been described as weak and transient[77]. To stabilise transient

interactions, we resorted to mild protein-protein crosslinking and treated cells with formaldehyde prior to lysate preparation (0.01% formaldehyde for 10 min). Based on the results of the ChIP-seq experiments, we anticipated co-purification of the transcription machinery along with the major co-transcriptional RNA processing complexes, including capping factors, the spliceosome, and the 3′ end processing machinery. We chose a protein that is known to reside in the same compartment (Srp2) as a reference rather than an untagged strain (Suppl. Fig. 2A). In our experience, untagged controls are not as informative because they tend to contain the same set of highly abundant cytoplasmic proteins. Co-purifying proteins were quantified by shotgun proteomics. In agreement with the ChIP-seq results, Dbp2-HTP and Srp2-HTP co-purified components of the RNAPII holoenzyme complex; amounts of co-purifying RNAPII were similar in both cases (Fig. 2A and Suppl. Fig. 2B). A notable exception was Rpb8, which co-

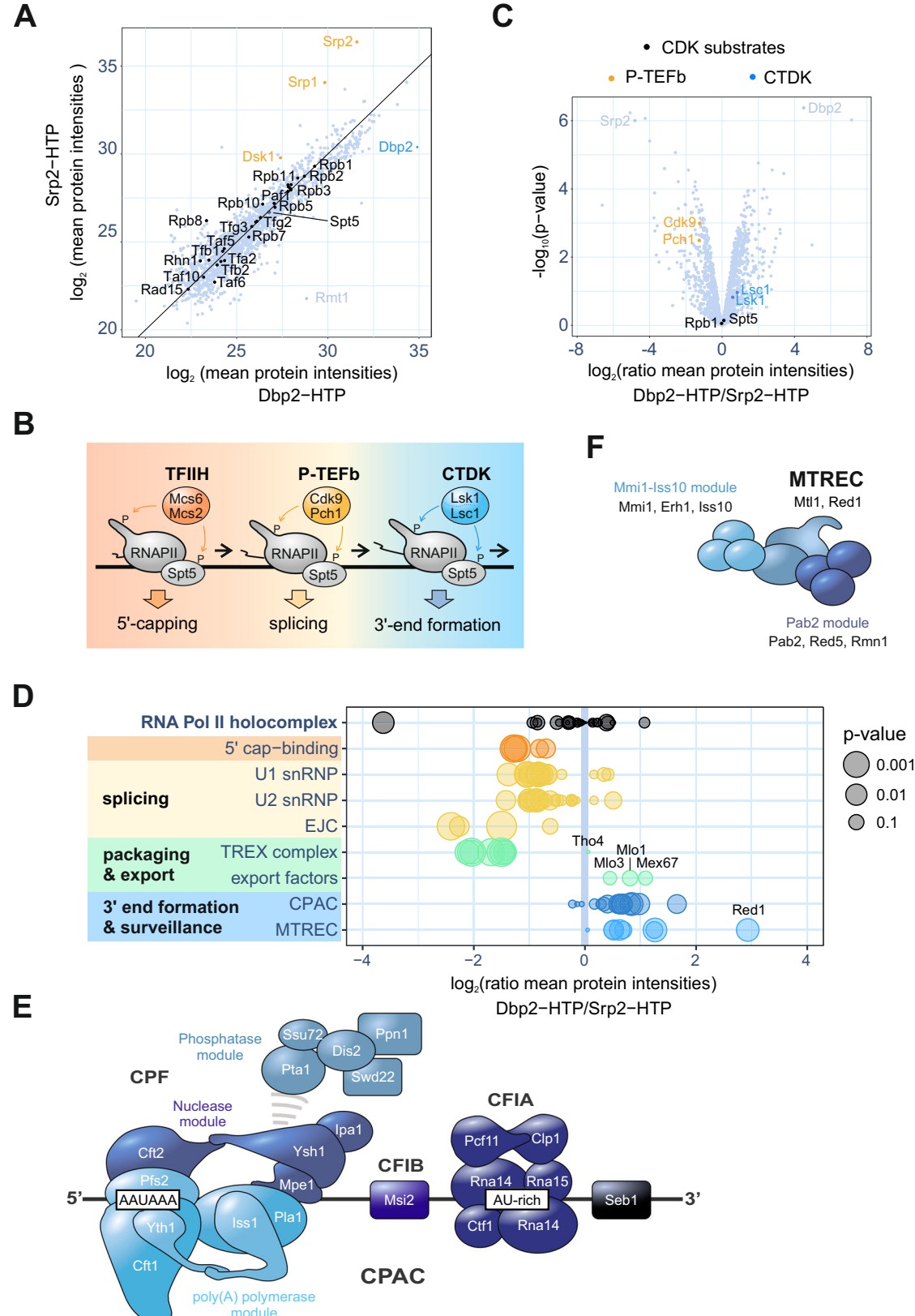

purified more readily with Srp2-HTP, potentially indicating a direct protein-protein interaction; the results of interaction modelling with AlphaFold2 multimer are consistent with this possibility (Suppl. Fig. 2C). Known interactors of Srp2, including Srp1 and the SR protein kinase Dsk1[78], were significantly enriched in the Srp2-HTP purification (Fig. 2A and Suppl. Fig. 2B). The most highly enriched protein in the

purification of Dbp2-HTP was the type I protein arginine N-methyltransferase (PRMT1) Rmt1 (Fig. 2A and Suppl. Fig. 2B). PRMT1 proteins have a strong preference for methylating RGG/RG motifs, multiple copies of which are present in the C-terminus of Dbp2, and the human Dbp2 orthologue DDX5 is a known target of arginine methylation[79,80].

**Fig. 2 | Comparative protein interaction profiling of Dbp2 and Srp2. A** Mass spectrometry (MS) analysis of the comparative interaction profiling of Dbp2 and Srp2 (n = 3 biological replicates). Components of the RNAPII holocomplex (GO:0016591) and the elongation factor Spt5 are marked in black, Srp2 and known interactors in orange, and Dbp2 in light blue. Source data are provided as a Source Data file. **B** Schematic of the CDK-dependent transcription cycle in fission yeast. In the presence of its cognate cyclin Mcs2, the TFIIH-associated CDK Mcs6 at the promoter phosphorylates Spt5 as well as RNAPII (at S5 of the CTD) to recruit capping factors. The activity of P-TEFb and its associated CDK/cyclin pair Cdk9/Pch1 is linked to promoter release and splicing. A second S2-specific CDK complex, CTDK, promotes cleavage and polyadenylation and transcription termination at the 3' end of genes[76]. **C** Relative enrichment of cyclin-dependent kinase (CDK) complexes and CDK substrates co-purifying with Dbp2 and Srp2. In the volcano plot, p-values (−log10, moderated two-sided Student's t-test) are plotted against the relative enrichment of proteins in the purification of Dbp2-HTP relative to Srp2-HTP based on mean protein intensities (log₂) (n = 3 biological replicates). The P-TEFb- and CTDK-associated CDK/cyclin pairs are marked in yellow and blue, respectively. The CDK substrates Rpb1 and Spt5 are marked in black. Source data are provided as a Source Data file. **D** Relative enrichment of RNA processing complexes co-purifying with Dbp2 and Srp2 (n = 3 biological replicates). The size of the circles reflects p-values (moderated two-sided Student's t-test). snRNP – small nuclear ribonucleoprotein. EJC – exon junction complex. CPAC – mRNA cleavage factor complex. MTREC – Mtl1-Red1 core. Volcano plots with the individual proteins labelled are provided in Suppl. Fig. 2D, E. Source data are provided as a Source Data file. **E** Schematic of the *S. pombe* cleavage and polyadenylation complex (CPAC), consisting of cleavage and polyadenylation factor (CPF), cleavage factors IA and IB (CFIA and CFIB) and the termination factor Seb1[29]. **F** Schematic of the *S. pombe* nuclear exosome targeting factor MTREC (Mtl1/Red1 core) with its associated modules[29].

Next, we analysed the relative enrichment of RNA biogenesis factors. The 5' to 3' shift in the chromatin association patterns of Srp2-HTP and Dbp2-HTP had already suggested that the proteins interact with RNAPII in different stages of the transcription cycle (Fig. 1C). This is further supported by the comparative interaction profiling data: The cyclin-dependent kinases (CDKs) placing the phospho-marks on RNAPII and Spt5 are recruited in a sequential manner[76] (Fig. 2B). Both CDK substrates are abundantly detected in both purifications (Fig. 2A, C). P-TEFb (Cdk9/Pch1 in *S. pombe*), an early RNAPII-S2P kinase with strong links to pre-mRNA splicing, was significantly enriched in the Srp2-HTP purification. In contrast, the late RNAPII-S2P kinase complex CTDK (Lsk1/Lsc1) preferentially co-purified with Dbp2-HTP, although the enrichment was not significant (Fig. 2C). Only very low levels of the initiation factor TFIIH were detected in both purifications, with the associated S5P kinase complex Mcs6/Mcs2 not co-purifying at all. We conclude that Srp2 and Dbp2 are recruited to transcribe RNAPII after promoter clearance. This is consistent with the ChIP-seq profiles, where chromatin-associated Srp2 and Dbp2 show little overlap with the promoter-associated RNAPII peak (Fig. 1C).

### Dbp2 preferentially associates with 3'-end formation factors and the nuclear RNA surveillance machinery

In agreement with the function of Srp2 as a splicing regulator, splicing factors were enriched in the Srp2-HTP purification, as were cap-associated proteins and the exon junction complex (EJC; Fig. 2D and Suppl. Fig. 2D, E). The TREX complex, an RNA packaging complex that couples mRNA transcription, processing and nuclear export[81,82], was significantly enriched in the Srp2-HTP purification, with the exception of Tho4, the RNA export adaptor homologous to the human TREX component ALYREF/THOC4 (Suppl. Fig. 2E). 3'-end processing factors preferentially co-purified with Dbp2-HTP, in agreement with its presence during the late steps of the transcription cycle. An interaction of Dbp2 with CFIA components Rna15 and Pcf11 and CFIB component Msi2 could be validated by non-crosslinking co-immunoprecipitation, although the interaction with Pcf11 was very weak (Fig. 2E and Suppl. Fig. 2F). In addition, several components of the exosome targeting factor MTREC and RSC, a SWI/SNF family chromatin remodelling complex[83,84], were significantly enriched in the Dbp2-HTP purification (Fig. 2D and Suppl. Fig. 2E–G).

### Dbp2 localises to cleavage bodies

Red1, a core component of MTREC[85–87], was among the most highly enriched proteins in the Dbp2-HTP purification (Fig. 2F and Suppl. Fig. 2E). MTREC is not a general component of the transcription machinery but is specifically recruited to exosome target genes by the YTH-domain protein Mmi1, which recognises TNAAAC motifs on nascent RNA and primes transcripts for decay by the nuclear exosome[67,88–90]. We observed no preferential recruitment of Dbp2 to Mmi1-regulated genes (Suppl. Fig. 3A). However, MTREC and CPAC components co-localise to discrete nuclear bodies in mitotically dividing fission yeast[27,85,91–94]. There is a compositional overlap between these nuclear exosome foci and the cleavage bodies described in human cells[91,95,96]. For the sake of brevity, we will refer to MTREC-containing foci as cleavage bodies throughout this work. Having found Dbp2 purifications to be enriched for many proteins associated with cleavage bodies in *S. pombe*, we asked whether Dbp2 was also present in these granules. A genomic C-terminal fusion of Dbp2 to GFP has been reported to localise to the nucleolus[97], which we also observed. In addition, Dbp2-GFP localises to punctate structures in the nucleus, which are often adjacent to the nucleolus or the nuclear rim and colocalize with Red1-tdTomato (Fig. 3A). The subcellular distribution of Dbp2-GFP strongly resembles that of the nuclear exosome component Rrp6 but differs from the CPAC components Rna15-GFP and Msi2-GFP or the nuclear poly(A)-binding protein Pab2, which are excluded from the nucleolus and dispersed throughout the nucleoplasm (Fig. 3A). Importantly, a common feature of all these factors is their colocalisation with Red1-tdTomato in foci (Fig. 3A, merged panel), suggesting that they are present in the same subnuclear structure. Imaging of a Pab2-tdTomato Dbp2-GFP strain confirmed that although Dbp2 and Pab2 are predominantly localised in different, non-overlapping compartments (nucleolus and nucleoplasm, respectively), they intersect in cleavage bodies that frequently occur at the boundary between both compartments (Fig. 3B, C).

### Cleavage bodies may help to buffer CPAC levels

Because 3'-end processing occurs co-transcriptionally[9,15], we wondered whether cleavage bodies represent highly active sites of transcription. We carried out immunofluorescence microscopy with an antibody against S2P-modified Rpb1 to detect transcriptionally active RNAPII in a strain where the MTREC component Mtl1 was C-terminally tagged with GFP as a marker for cleavage bodies. Rpb1-S2P yielded a granular staining that overlapped with the DNA-rich nucleoplasmic compartment counterstained with DAPI (Suppl. Fig. 3B). While cleavage bodies were frequently detected adjacent to an area of higher Rpb1-S2P density, this was largely limited to the surface of the RNAPII transcriptionally active compartment, particularly at the inner arc of the crescent shape corresponding to the interface with the nucleolus. Overall, the majority of RNAPII transcription hot spots were not associated with a cleavage body, and Mtl1-GFP foci did not usually coincide with a peak in Rpb1-S2P signal (Suppl. Fig. 3B, C).

We deduce that cleavage bodies, while containing components of the 3'-end processing machinery, are unlikely to represent the sites where the bulk of RNA cleavage and polyadenylation occurs. Therefore, we considered whether cleavage bodies serve as storage compartments for 3'-end processing factors. To test this hypothesis, we analysed an analogue-sensitive mutant of Cdk9, *cdk9-as*, which exhibits promoter-proximal stalling of RNAPII and elongation defects within minutes after addition of the bulky purine analogue 3-MP-PP1[17,98].

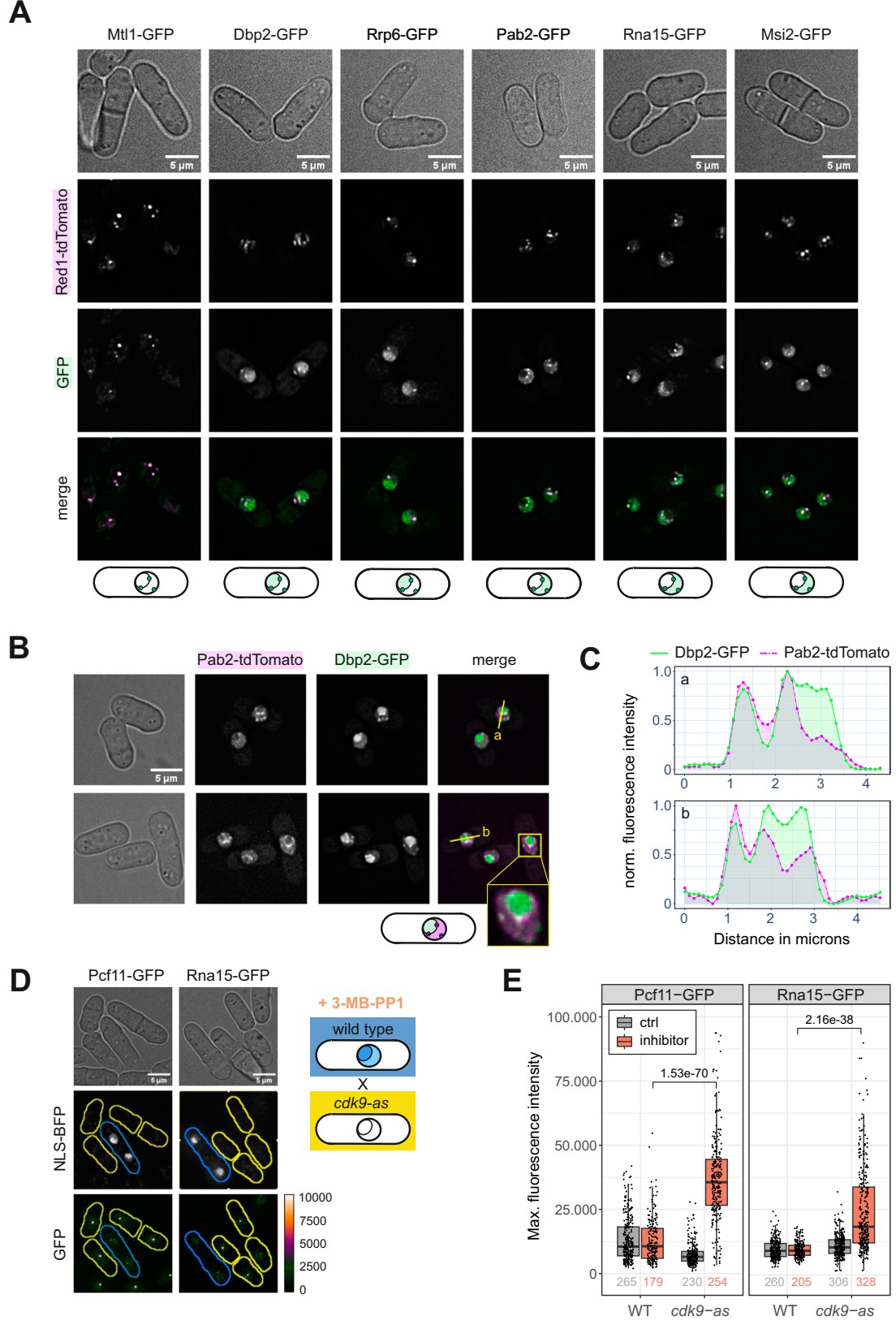

In the presence of 3-MP-PP1, fluorescent signal of Pcf11-GFP and Rna15-GFP in foci is significantly increased in the analogue-sensitive strain, but not in the wild-type control (Fig. 3D, E and Suppl. Fig. 3D, E), supporting the idea that cleavage bodies are storage sites and may help to buffer CPAC levels. However, we cannot exclude that CPAC components within cleavage bodies contribute to processing of mRNAs that have

been prematurely released or carry out re-cleavage of polyadenylated transcripts[99].

**In the absence of Dbp2, poly(A) + RNA is retained in the nucleus**
Our data indicated that Dbp2 interacts with the 3′-end processing machinery and RNA export factors, suggesting that it might bridge

**Fig. 3 | Dbp2 is a component of cleavage bodies. A** Live cell imaging of GFP-tagged RNA processing factors and the MTREC component Red1-tdTomato as a marker for cleavage bodies. The merged channel shows GFP in green and Red1-tdTomato in magenta. The sketch below indicates the typical localisation observed for the GFP-tagged protein in at least three independent experiments. **B** Live cell imaging of genomically tagged Dbp2-GFP and the nuclear poly(A)-binding protein Pab2-tdTomato. Images are representative of three independent experiments. Fluorescence intensity profiles were generated along the yellow line. **C** Fluorescence intensity profiles along the lines indicated in B. Green line corresponds to the Dbp2-GFP signal, dashed magenta line to the Pab2-tdTomato signal. Fluorescence intensities were normalised to a 0-1 range. Source data are provided as a Source Data file. **D** Live cell imaging of mixed cultures of wild type (marked by NLS-BFP; blue outlines) and an analogue-sensitive mutant of Cdk9 (*cdk9-as*; yellow outlines) with either Pcf11 or Rna15 tagged with GFP. Cells were grown in in YES at

30 °C and treated with 100 µM 3-MB-PP1 for 10 min to block transcription elongation before pelleting and resuspension in EMMG + 3-MB-PP1 for imaging. The GFP channel is shown as the maximal intensity projection using a blue orange icb colour scale as indicated. Images for the untreated controls are provided in Suppl. Fig. 3D, E. **E** Quantification of the inhibitor experiment shown in (**D**) (*n* = 2 independent experiments). Measurements were performed on maximal intensity Z-projections and the maximal fluorescence intensity scored for each cell. Non-treated mixed cultures were included as controls. The numbers below the graph indicate the total number of counted cells for each condition. The lower and upper hinges correspond to the first and third quartiles, and the whiskers extend from the hinge to the smallest and largest value no further than 1.5 * IQR from the hinge (where IQR is the inter-quartile range). The horizontal line marks the median value. The displayed p-values for the pair-wise comparisons were calculated using a two-sided Wilcoxon rank sum test. Source data are provided as a Source Data file.

these two processes. We hypothesised that Dbp2 – as a DEAD-box ATPase and potential RNPase – could be involved in CPAC release from processed transcripts as part of the mRNP assembly checkpoint[36]. Because Dbp2 was resistant to depletion via an auxin-dependent degron (Song et al.[100] and own observation), we placed the endogenous *dbp2* gene under the control of the *nmt1* promoter, which is repressed in the presence of thiamine[101]. We used a Myc-tagged strain to monitor the kinetics of protein depletion after a shift to thiamine-containing medium by Western blot (Fig. 4A, B).

To determine the localisation of RNAs that have undergone processing by cleavage and polyadenylation, we performed poly(A) + RNA FISH. As a control, we included a heat shock treatment, which is known to efficiently block bulk mRNA export in yeast[102]. In wild-type cells, poly(A) + RNA is evenly distributed throughout the cytoplasm and moderately more abundant in the nucleus, with the mean ratio of average fluorescence intensities of nucleus and cytoplasm at 1.34 (Fig. 4C, D). Upon heat shock, we observed almost complete nuclear retention of poly(A) + RNA, which accumulates in a striking ring-shaped pattern from which chromatin appears to be excluded. These poly(A) + RNA-rich structures are likely to correspond to nucleolar rings, reversible aggregates of nuclear RNA metabolism factors that form during heat stress[103]. Upon depletion of Dbp2, poly(A) + RNA is still present in the cytoplasm; however, we observed a significant increase in nuclear to cytoplasmic signal ratio (1.79; *p*-value = 36.5e-15 (two-sided Wilcoxon rank sum test)). We conclude that 3′-processed, polyadenylated RNAs are inefficiently exported into the cytoplasm in the absence of Dbp2.

Although much of the poly(A) + RNA that was retained in the nucleus in the *P.nmt-dbp2* mutant was dispersed in the nucleoplasm, we noted that the signal partially aggregated in foci. As we had previously determined that Dbp2 localises to cleavage bodies in wild-type cells, we asked whether these would be the sites where poly(A) + RNA accumulated in its absence. Using oligo-d(T) FISH in *P.nmt-dbp2* cells in which the cleavage body component Mtl1 was C-terminally tagged with GFP, we confirmed that the sites of poly(A) + RNA accumulation correspond to cleavage bodies, suggesting that Dbp2 is important for the release of poly(A)+ RNAs from this compartment (Suppl. Fig. 4A, B).

### MTREC components Red1 and Pab2 are not required for the nuclear retention of poly(A) + RNA in the absence of Dbp2

Recent work in human cells has identified the Red1 homologue ZFC3H1 as a nuclear retention factor for exosome target RNAs that can sequester RNA in nuclear condensates in a manner dependent on the nuclear poly(A)-binding protein PABPN1[104–107]. To test the involvement of MTREC in poly(A) + RNA retention in the *dbp2* mutant, we crossed *red1Δ*, *pab2Δ* (homologue of PABPN1) and *iss10Δ* into the *P.nmt-dbp2* background. Iss10 is a fission yeast-specific component of MTREC, which shows high homology to the N-terminal region of ZFC3H1 and is required for cleavage body integrity[26,85,108]. Combining *P.nmt-dbp2* with *red1Δ* or *pab2Δ* did not improve growth on YES; rather, it made it

slightly worse (Suppl. Fig. 4C). In contrast, combination with *iss10Δ* provided *P.nmt-dbp2* with a mild but reproducible growth advantage under depletion conditions; however, colony size was also slightly increased for *iss10Δ* alone compared to the wild type (Suppl. Fig. 4C).

To determine the impact of the additional mutations on poly(A) + RNA export, we carried out poly(A) + RNA FISH. The single deletion of *red1Δ* or *pab2Δ* led to a significant retention of poly(A) + RNA in foci even in the presence of Dbp2, with an additive effect on export if both mutations were combined (Suppl. Fig. 4D, E). The single *iss10Δ* deletion, on the other hand, had a very mild retention phenotype, but was equally unable to rescue nuclear poly(A) + RNA retention if combined with *P.nmt-dbp2* (Suppl. Fig. 4E). In addition, Dbp2-GFP was still present in cleavage bodies when either Red1 or Iss10 were deleted (Suppl. Fig. 4F). We conclude that additional, unidentified factors contribute to RNA retention in the *P.nmt-dbp2* background.

### CPAC components are depleted from cleavage bodies upon loss of Dbp2

If Dbp2 played an active role in CPAC release in the context of an mRNP assembly checkpoint, its depletion would be expected to lead to increased CPAC retention. To determine whether Dbp2 was involved in CPAC recycling, we first examined the subcellular distribution of CFIA components in the presence and absence of Dbp2 (Fig. 5A, B and Suppl. Fig. 5A, B). Upon depletion of Dbp2, fluorescent signal of Pcf11-GFP and Rna15-GFP in foci was significantly reduced. In wild-type cells, the majority of GFP-tagged Pcf11 colocalised with Red1-tdTomato in cleavage bodies, with little fluorescence detectable in the nucleoplasm (Fig. 5C, upper panel). In the absence of Dbp2, colocalisation with cleavage bodies was markedly reduced, and the fluorescent signal redistributed to the RNAPII-positive compartment (Rpb9-iRFP; Fig. 5C, lower panel, and 5E). Based on our earlier results, we assume that this reflects a redistribution from the storage compartment to the sites of RNA biogenesis.

### CPAC components are depleted from the soluble pool upon loss of Dbp2

We had noted earlier that lysates prepared from the *P.nmt-dbp2* strain appeared to contain much lower levels of various CPAC components than the wild type. However, this disagreed with fluorescence microscopy data, where we observed no change in the total fluorescent signal of GFP-tagged CPAC components between wild type and mutant. It has been observed previously that stalling of RNP complexes on chromatin can interfere with their efficient extraction using standard protocols for chromatin preparation[109]. We therefore compared protein amounts in standard SDS lysates with extracts prepared using the TCA method, which has been reported to be more efficient in extracting chromatin[110]. Indeed, while Dbp2 depletion strongly reduced the amounts of Pcf11 or of the CFIB component Msi2 recovered after SDS lysis, their yield after TCA lysis was not affected (Fig. 6A). This suggested that CPAC components are depleted from the

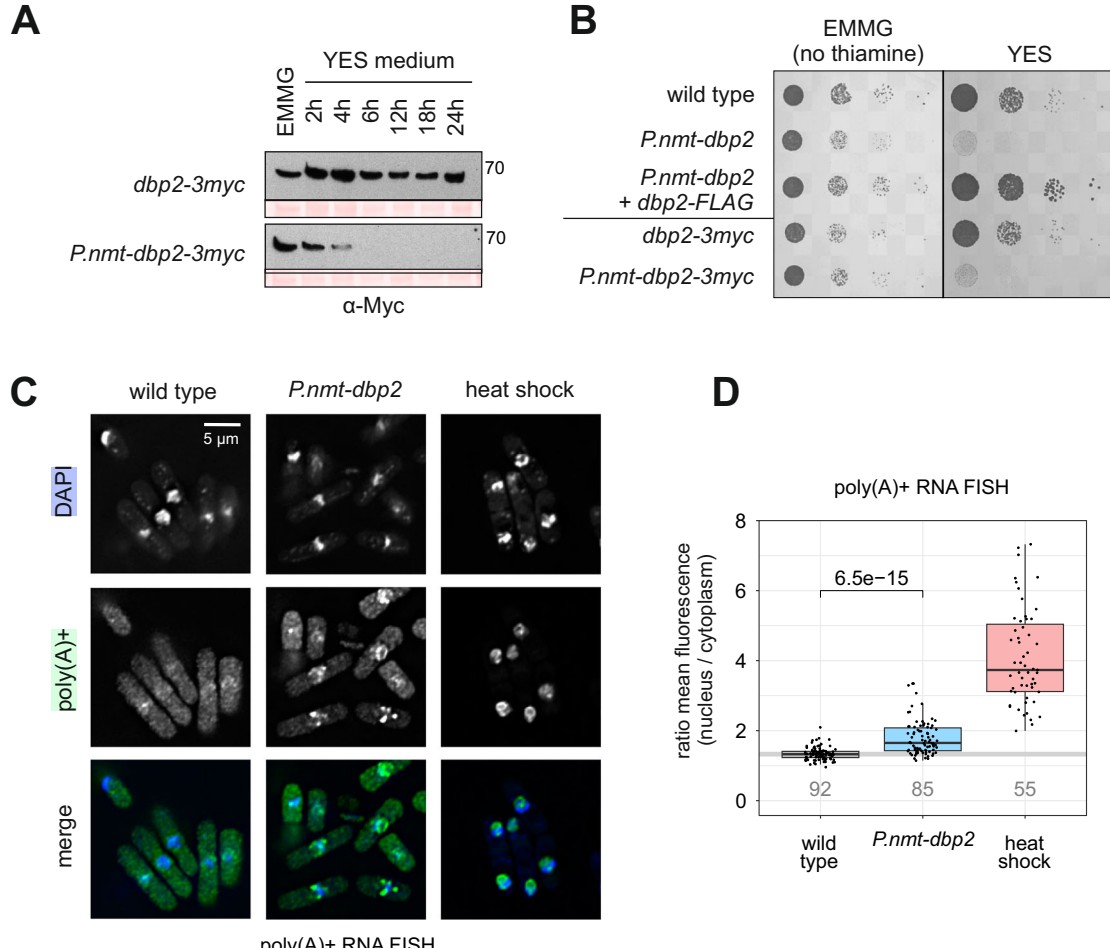

**Fig. 4 | In the absence of Dbp2, 3'-processed RNAs accumulate in the nucleus.**
**A** Western blot depletion time course for Dbp2-3myc under its endogenous or the *nmt1* promoter. Cells were grown in EMMG overnight at 30 °C and shifted to YES medium for the indicated amount of time. The numbers on the right indicate the molecular weight marker in kDa. Ponceau stained membrane is included as a loading control. The experiment was carried out independently three times, with similar results. **B** In a plate-based growth assay, metabolic depletion of Dbp2 or Dbp2-3myc under the *nmt1* promoter leads to poor growth on thiamine-containing medium (YES). The growth phenotype is rescued when an additional copy of *dbp2-FLAG* under its endogenous promoter is inserted at an ectopic locus (*leu1*). The indicated strains were grown in EMMG overnight and serial dilutions (1:10) were spotted on EMMG or YES and incubated at 30 °C. The image is representative of three independent experiments. **C** Fluorescence in-situ hybridisation (FISH) against poly(A) + RNA using a Cy3-labelled oligo-d(T) probe and DAPI to stain the DNA.

Cells were grown overnight in EMMG, then shifted to YES for 5 h to shut off the *nmt1* promoter before formaldehyde fixation. For the heat shock control, wild-type cells were shifted to 42 °C for 1 h. The merged channel shows poly(A) + RNA in green and DAPI in blue. **D** Quantitation of the oligo-d(T) FISH experiment in C (*n* = 3 independent experiments for wt and *P.nmt-dbp2*; *n* = 2 for heat shock). Measurements were performed on average intensity Z-projections for the nucleus and cytoplasm and the ratio of mean nuclear over mean cytoplasmic fluorescence intensity was calculated for each cell. The numbers below the graph indicate the total number of counted cells. Box plot features as in Fig. 3E. The displayed p-values for the pairwise comparisons were calculated using a two-sided Wilcoxon rank sum test. Note that the calculated ratios underestimate nuclear RNA retention because segmentation on the DAPI channel is biased towards the DNA-rich nuclear compartment and can exclude parts of the nucleolus. Source data are provided as a Source Data file.

soluble pool in the absence of Dbp2, possibly because they are sequestered on poly(A) + RNA retained in the nucleus and/or on chromatin. To differentiate between these possibilities, we treated crude SDS lysates with RNases or DNase before clearing the lysate by centrifugation. Treatment with DNase but not RNase was able to release Msi2-GFP into the soluble fraction in the *dbp2* mutant (Fig. 6B, C), suggesting that the protein is retained on chromatin. The same experiment for Pcf11-GFP was inconclusive because incubation of the crude lysate led to the recovery of soluble Pcf11 even in the absence of enzyme (Suppl. Fig. 6A).

**Dbp2 depletion increases skipping of cleavage and polyadenylation sites**
Assuming that CPAC components in the soluble pool correspond to the fraction that is available for the 3'-end processing of nascent transcripts, we wondered whether reduced levels of soluble CPAC in

the *dbp2* mutant would lead to inefficient cleavage and polyadenylation. We performed RNA-seq analysis on poly(A)-selected RNA to determine whether depletion of Dbp2 had an impact on PAS usage. We used *S. cerevisiae* cells as a spike-in for normalisation to be able to assess the global impact on mRNA expression levels. We observed a transcriptome-wide reduction in RNA levels after Dbp2 depletion, in agreement with an essential function for Dbp2 as a global regulator of RNAPII-dependent RNA biogenesis (Suppl. Fig. 7A, B). Similar results were obtained when we sequenced ribodepleted RNA after a 9 h depletion period (Suppl. Fig. 7B). Because Dbp2 co-purified and co-localised with MTREC, we also verified whether decay of exosome substrates was affected. Levels of known MTREC-dependent RNA targets of the nuclear exosome, including meiotic mRNAs that are part of the Mmi1 regulon and snoRNA precursors that depend on the nuclear membrane protein Lem2 for decay[25,67,111], were not increased relative to other transcripts upon depletion of Dbp2 (Suppl. Fig. 7C)

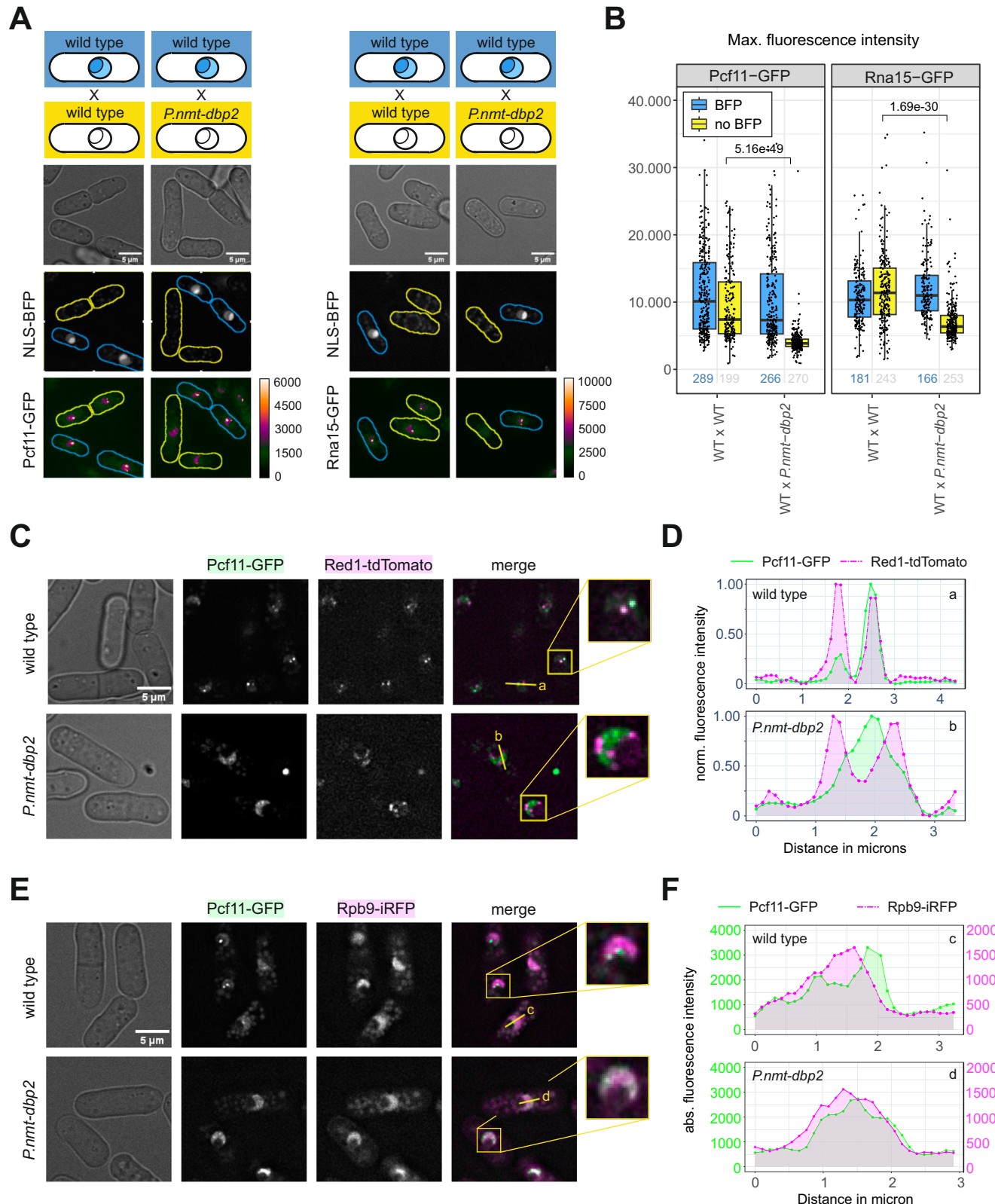

indicating that Dbp2 is not required for MTREC-dependent RNA turnover.

For many mRNAs, we observed a relative increase in sequencing reads over longer 3'-UTR isoforms or past the CPA site of annotated features after Dbp2 depletion, suggesting a general tendency of the mutant to skip sites and continue transcription to downstream sites

(Fig. 7A, B and Suppl. Fig. 7D). In the RNA-seq data, increased read-through signal could be detected for almost any transcript with a sufficiently high expression level; however, the severity of the phenotype was transcript-dependent and may be related to the strength of the corresponding PAS. Northern blotting with probes directed against the gene bodies confirmed the presence of extended transcript

**Fig. 5 | CPAC components are depleted from cleavage bodies and redistribute to the RNAPII compartment in the absence of Dbp2. A** Live cell imaging for Pcf11-GFP or Rna15-GFP in wild type (marked by NLS-BFP) mixed with *P.nmt-dbp2*. Cells were shifted to YES for 6 h. The GFP channel is shown as the maximal intensity projection using a blue orange icb colour scale as indicated. Mixed wild type (+/-BFP) was included as control. Extended fields of view are provided in Suppl. Fig. 5A, B. **B** Quantification of the experiment shown in (**A**) (*n* = 2 independent experiments). Measurements were performed on maximal intensity Z-projections and the maximal green fluorescence intensity scored for each cell. Mixed cultures of wild types were included as controls. Numbers below the graph indicate number of counted cells for each condition. Box plot features as in Fig. 3E. The displayed p-values for pair-wise comparisons were calculated using a two-sided Wilcoxon rank sum test. Source data are provided as a Source Data file. **C** Live cell imaging of genomically tagged Pcf11-GFP and the MTREC component Red1-tdTomato. Cells

were shifted to YES for 6 h. The merged channel shows Pcf11-GFP in green and Red1-tdTomato in magenta. Images are representative of three independent experiments. Fluorescence intensity profiles along the yellow lines are shown in (**D**). **D** Fluorescence intensity profiles as indicated in (**C**). Green line corresponds to the Pcf11-GFP signal, dashed magenta line to the Red1-tdTomato signal. Fluorescence intensities were normalised to a 0–1 range. Source data are provided as a Source Data file. **E** Live cell imaging of genomically tagged Pcf11-GFP and the RNAPII component Rpb9-iRFP. Cells were shifted to YES for 6 h. The merged channel shows Pcf11-GFP in green and Rpb9-iRFP in magenta. Images are representative of two independent experiments. Fluorescence intensity profiles along the yellow lines are shown in **F**. **F** Fluorescence intensity profiles as indicated in (**E**). Green line corresponds to the Pcf11-GFP signal, dashed magenta line to the Rpb9-iRFP signal. Source data are provided as a Source Data file.

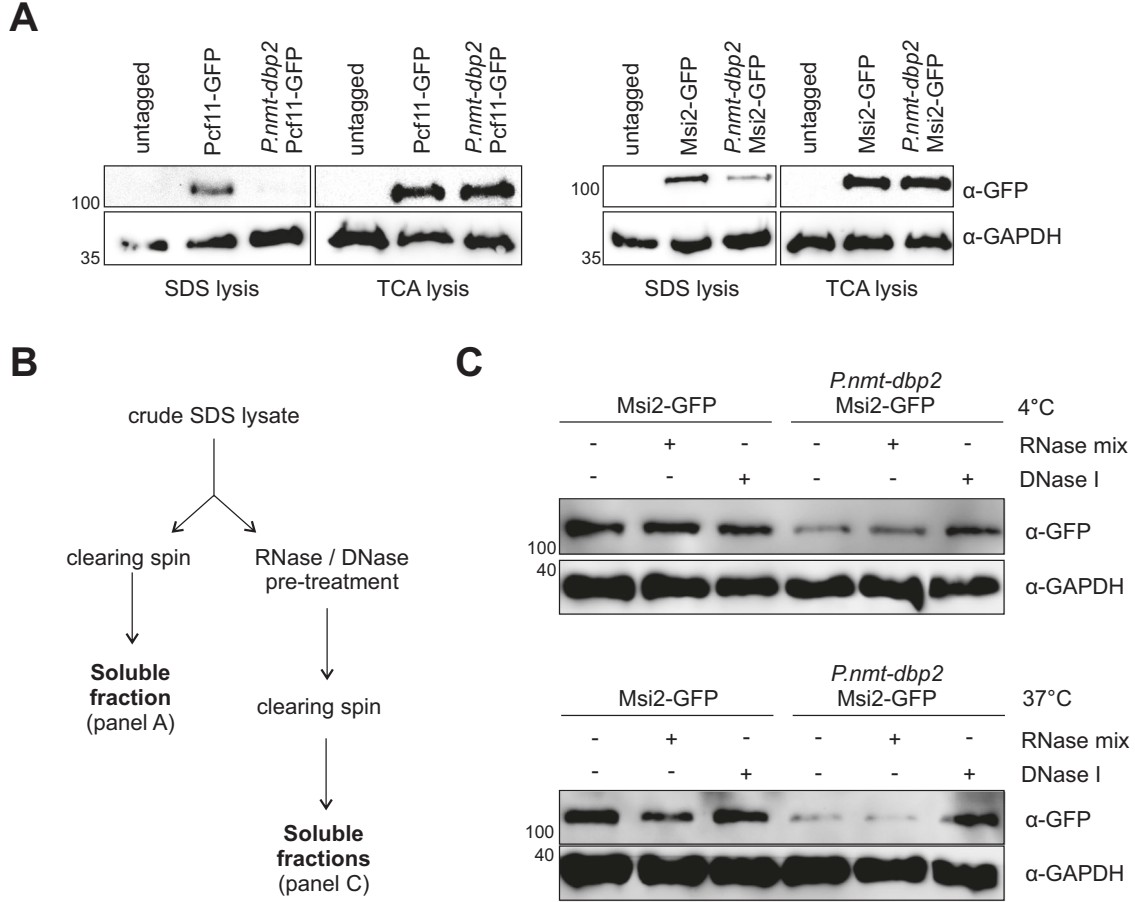

**Fig. 6 | CPAC components are depleted from the soluble pool in the absence of Dbp2. A** Western blot analysis of cell extracts generated by SDS lysis (soluble cell extracts) or TCA lysis (total cell extracts) for CFIA component Pcf11-GFP (left panels) and CFIB component Msi2-GFP (right panels). Cells were grown in EMMG overnight at 30 °C and shifted to YES for 5 h before lysis. GAPDH was detected on the same blot and is included as a loading control. The numbers on the left indicate the molecular weight marker in kDa. Images are representative of two independent

experiments. **B** Schematic of SDS lysate generation in A and C. **C** Western blot analysis of SDS lysates incubated with RNases or DNase for either 30 min at 4 °C (top panel) or 15 min at 37 °C (bottom panel) prior to the clearing spin to release RNA- or DNA-associated proteins into the soluble pool. Lysates incubated without added enzymes were included as control. Images are representative of two independent experiments.

isoforms for several mRNAs that we tested (Suppl. Fig. 7E). For *rpl2501* mRNA, we included a probe against the extended 3'-UTR to validate the increased presence of 3'-extended transcripts in the mutant (Fig. 7C). Moreover, analysis of CPA site usage on this transcript by RT-PCR amplification using an anchored (T)$_{12}$VN primer confirmed that Dbp2 depletion leads to alternative PAS usage and 3'UTR lengthening. This phenotype is typically observed upon depletion of core poly(A) factors[112], suggesting that one or several CPAC components may indeed become limiting in the *dbp2* mutant.

## In the absence of Dbp2, transcription termination is delayed

Because transcription termination is coupled to co-transcriptional RNA cleavage, we also sought to determine the impact of Dbp2 depletion on RNAPII transcription. Using spike-in controlled RNAPII-ChIP-seq, we observed a global reduction of RNAPII levels at transcribed genes upon depletion of Dbp2 (Fig. 8A, B). To determine whether transcription termination was affected, we performed meta-gene analysis of RNAPII coverage at RPGs. In the absence of Dbp2, the peak in RNAPII coverage at the 3'-end of genes shifted downstream

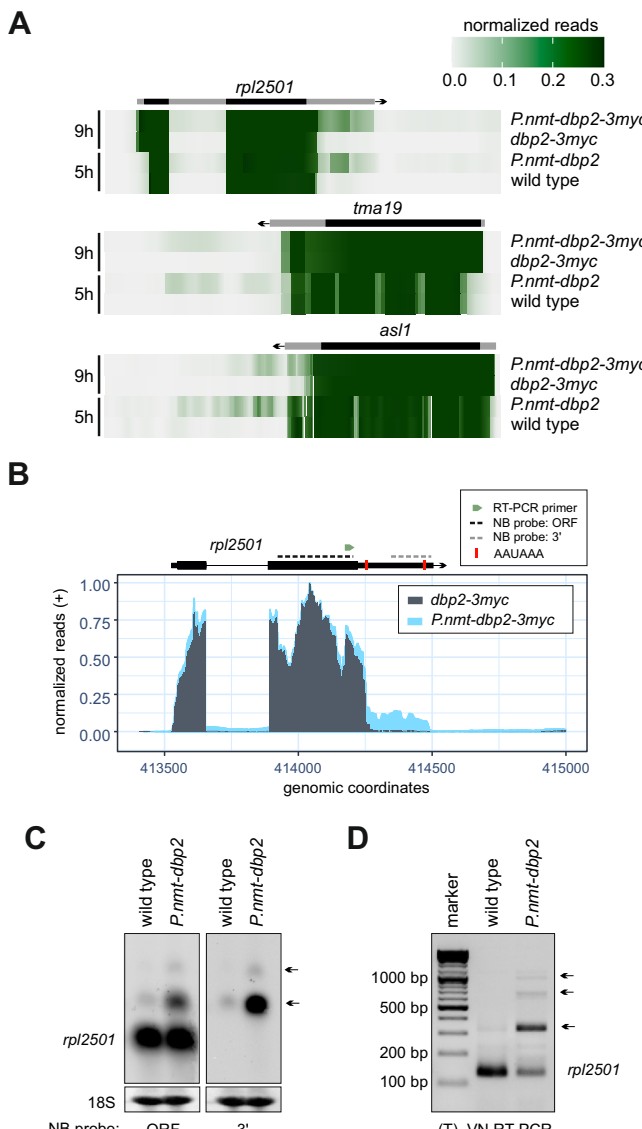

**Fig. 7 | Dbp2 depletion increases skipping of cleavage and polyadenylation sites. A** Heat maps of RNA-seq reads across *rpl2501*, *tma19*, and *asl1*, including downstream regions in untagged (n = 3 biological replicates) or 3myc-tagged Dbp2 (n = 2 biological replicates) after a 5 h or 9 h depletion period, respectively. Reads were counted across 10 bp bins, the means normalised to a 0–1 range and plotted at $y_{max}$ = 0.3 to visualise relative amounts of 3′-extended transcripts. The positions of the annotated transcripts are indicated in grey, the coding sequence is highlighted in black. **B** Representative RNA-seq traces (positive strand) across the region of chromosome II encompassing the *rpl2501* gene with reads for *dbp2-3myc* in grey, for *P.nmt-dbp2-3myc* in light blue. Read counts were normalised to a 0–1 range. Cells were grown in EMMG overnight at 30 °C and shifted to YES medium for 9 h prior to RNA isolation and ribodepletion. The position of polyadenylation signals (AAUAAA) is indicated as red bars, the Northern probes used in C as dotted lines and the forward primer used for PCR amplification (**D**) as green arrowhead. **C** Northern blot for *rpl2501* mRNA using strand-specific DIG-labelled RNA probes against the gene body or the 3′-extension as indicated in (**B**). 18 S band stained with methylene blue is shown as loading control. Cells were grown in EMMG and shifted to YES medium for 6 h prior to RNA isolation. Arrows indicate positions of extended transcripts. **D** Analysis of poly(A) site usage by RT-PCR amplification using an anchored (T)$_{12}$VN primer. RNA was prepared as in (**C**). The (T)$_{12}$VN primer was used to prime the reverse transcription reaction before PCR amplification with a gene-specific forward primer (indicated in B) and an additional priming sequence contained within the RT primer for 30 cycles. Usage of the AAUAAAA sites marked in B is expected to lead to amplicons of ~140 bp and ~350 bp, respectively. Amplicons that reflect usage of distal poly(A) sites are marked with arrows. Images are representative of three independent experiments.

compared to the wild type, indicating a delay in transcription termination (Fig. 8C, left panel). According to the prevalent model of cleavage-coupled transcription termination, RNAPII is in kinetic competition with the 5′-3′ exonuclease Dhp1 (homologue of Xrn2), which degrades the nascent 3′-fragment that is generated during cleavage and promotes transcription termination once it reaches RNAPII[12,13,15,113]. Because of this competition, both faster RNAPII elongation rates and a delay in RNA cleavage can delay transcription termination. At present, we cannot unambiguously distinguish between these possibilities. However, relative to the amount of RNAPII that reaches the PAS, the height of the RNAPII peak across the termination zone is increased upon Dbp2 depletion, which is indicative of a slow transition of RNAPII through the area (Fig. 8C, right panel). We therefore consider it likely that depletion of Dbp2 decreases the efficiency of CPA at RNAPII-transcribed genes, which is consistent with the increased occurrence of PAS skipping that we observed on the RNA level. We take this as an indication that the depletion of CPAC components from the soluble pool in the absence of Dbp2 limits their availability to levels that are insufficient to maintain normal levels of 3′-end processing.

## Discussion

Despite an extensive body of work that has advanced our understanding of the biological function of the RNP remodelling helicase

Dbp2/DDX5, the mechanisms underlying the pleiotropic phenotypes associated with its dysfunction are still elusive. In this study, we provide evidence that places fission yeast Dbp2 at the intersection between RNA 3′-end formation and RNA export. Our data is consistent with a model in which Dbp2 is the key enzyme in an early mRNP remodelling checkpoint that is coupled to the release of CPAC into the soluble pool and allows 3′-end processed transcripts to gain export competence. Importantly, this would establish transcript release after 3′-end formation as an ATP-dependent process.

The existence of such a checkpoint has been proposed based on the analysis of RNA export mutants in *S. cerevisiae*[36]. Here, temperature-sensitive alleles of the export receptor Mex67, the export adaptor Yra1, and various other RNA assembly factors led to the retention of the CFIA components Rna14 and Rna15 on polyadenylated RNA. Conversely, some mutations in CFIA that retain the ability to support cleavage and polyadenylation lead to a failure to export the processed transcripts[114]. Notably, our comparative interaction profiling experiment identified several RNA export factors that associate with Dbp2, among them Mex67 and the Yra1 homologue Mlo3; in *S. cerevisiae*, *Sc*Mex67 and Yra1 have also been shown to interact with *Sc*Dbp2 and regulate its function, and deletion of *DBP2* – which is not an essential gene in budding yeast – was shown to reduce association of Mex67 with transcripts[115,116]. In *Drosophila*, a strong similarity between the phenotypes of mutants of the Mex67 and Dbp2 homologues *small bristles* and *Rm62* has been described[73]. An obvious hypothesis would be that direct interaction between export factors and Dbp2 initiates CPAC displacement. It will therefore be of great interest to investigate the mode of interaction between Dbp2 and these export factors in further studies.

At present, we can only speculate about the mechanism of RNA retention in the absence of Dbp2. A non-negligible amount of the nuclear poly(A) + RNA signal in *P.nmt-dbp2* is present in cleavage bodies (Suppl. Fig. 4A, B), suggesting that they might serve a function analogous to nuclear speckles (NS) in metazoans. NS are enriched in splicing factors and have been proposed to either serve as an active buffering compartment for splicing factors or to represent processing hubs[117]. They are also the sites where poly(A) + RNA accumulates upon an export block; therefore, passage through NS is thought to allow

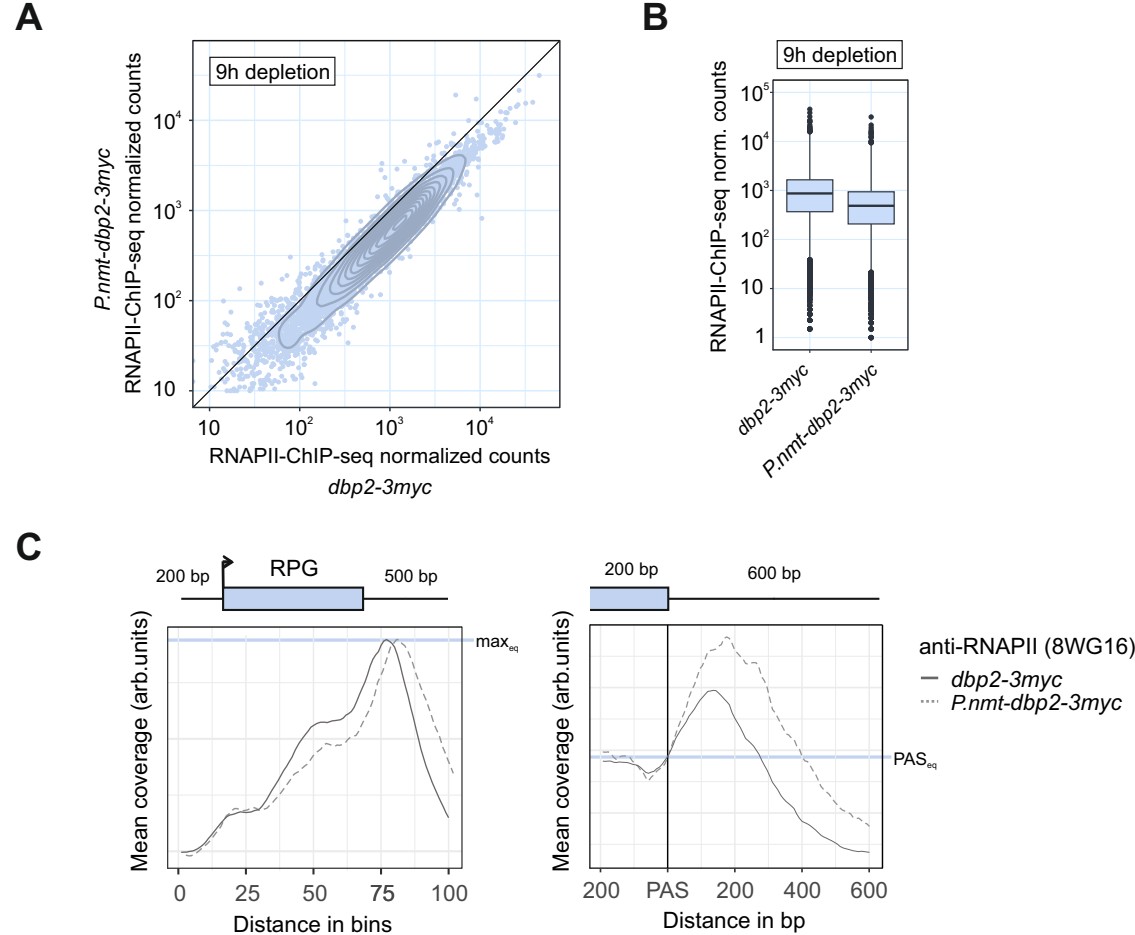

**Fig. 8 | In the absence of Dbp2, transcription termination is delayed. A** Mean integrated counts of RNAPII-ChIP-seq reads over annotated features in *dbp2-3myc* or *P.nmt-dbp2-3myc* ($n = 2$ independent experiments) after a 9 h depletion period. Cells were grown in EMMG and shifted to YES for 9 h, then mixed with *S. cerevisiae* in a 5:1 $OD_{600}$ ratio prior to chromatin isolation and IP against total RNAPII (α-rpb1 (8WG16)). Integrated counts were normalised to the number of total *S. cerevisiae* reads for each sample before calculating the mean. Contours of a 2d density estimate are shown in dark grey. Source data are provided as a Source Data file. **B** Mean integrated counts of RNAPII-ChIP-seq reads over annotated features as in A. Box plot features as in Fig. 3E. Outliers are plotted individually. **C** Metagene analysis of mean RNAPII ChIP-seq signal ($n = 2$ independent experiments) across ribosomal protein genes (RPGs) including 200 bp upstream and 500 bp downstream of the annotated transcription units (left panel) or surrounding the polyadenylation and cleavage site (PAS) (right panel). Mean coverage was adjusted by a constant scaling factor to normalise to maximal peak height (left panel) or RNAPII levels at the PAS (right panel) as indicated by the grey horizontal lines. Schematic of the gene above the left panel corresponds to an RPG of median length. In the left panel, the positions of transcription start and end sites within the metagene are distributed around the given coordinate because of the varying feature compression depending on gene length.

mRNAs to acquire export competence[118]. However, in contrast to *red1Δ* and *pab2Δ*, where nuclear-retained poly(A)+ RNAs almost exclusively localize to cleavage bodies, disperse poly(A) + RNA constitutes a significant fraction of the total nuclear signal after Dbp2 depletion (Fig. 4C and Suppl. Fig. 4D). At present, our best assumption is that RNA retention already occurs at an early stage of RNA biogenesis and may represent a failure to release RNA from chromatin after transcription. This interpretation is supported by live cell imaging of CPAC components and their depletion from the readily extractable soluble pool (Fig. 5A, C), which – in the case of Msi2 – is reversible by DNase treatment. In murine cells expressing a version of Rpb1 with a truncated CTD, fully processed mRNAs are retained at the transcribed gene, suggesting that release from chromatin constitutes an independent step of gene expression that requires an active component usually recruited by the full-length CTD[119]. Recent experiments that measured RNA residence times in various compartments in mammalian cells identified release from chromatin as the rate-limiting step for nuclear export[120]; a similar observation had been made earlier for a subclass of mRNAs encoding inflammatory proteins, and in that case had been linked to slow splicing kinetics[121]. Poor RNA processing has

long been known to result in the retention of RNAs at the site of transcription, which is supported by genome-wide correlation studies[109,122–124]. Whether this retention is mechanistically related to the RNA retention we observe after the loss of Dbp2 is currently unclear. However, in *S. cerevisiae*, the CFIB component and Msi2 homologue Hrp1 was recently shown to be required to prevent the export of incorrectly processed transcripts to the cytoplasm[125]. There is relatively little published work that has directly addressed the question of how selective tethering of RNA to chromatin can be achieved: For mutants of the TREX complex in *S. cerevisiae*, chromatin retention has been convincingly linked to the formation of R-loops[126]. In that case, retention coincides with downregulation of the CPAC component Fip1, a co-factor of the poly(A) polymerase, and defects in polyadenylation[127]. Whether the failure to polyadenylate the transcript is a prerequisite for strand invasion and efficient RNA:DNA hybrid formation in TREX mutants is an open question – if it was, it would be unlikely that the poly(A) + RNA that we observe in the absence of Dbp2 is tethered to chromatin by the same mechanism. As an alternative to direct tethering to DNA as part of an RNA:DNA hybrid, retention on chromatin could also be linked to the formation of RNA-rich protein

condensates. In addition to Dbp2 itself, which has the ability to induce phase separation[128], at least one 3'-end processing factor in *S. pombe*, the CTD-interacting protein Seb1[129,130], has been described to form chromosome-tethered condensates under specific conditions in a recent preprint[131]. Alternative mechanisms of RNA retention are conceivable, and contributing factors and the exact mechanism will have to be determined in the future.

### Links to the nuclear RNA surveillance machinery

Many processing defects that induce the retention of specific transcripts on chromatin are linked to the degradation of the retained transcript by the nuclear exosome[122,132,133]. Notably, we find that Dbp2 both co-purifies and co-localises with MTREC, the nuclear exosome targeting complex related to human PAXT composed of the Mtr4-like helicase Mtl1, the zinc-finger protein Red1, and various associated factors including the nuclear poly(A)-binding protein Pab2[26,85,86,134–136]. To date, we have found no evidence for a direct link between Dbp2 and MTREC function. Dbp2 is not required for the turnover of known exosome targets (Suppl. Fig. 7C), and RNA retention after Dbp2 depletion is not dependent on Red1 or Pab2 (Suppl. Fig. 4D, E). Rather, the effects of both mutations on RNA retention appear to be additive, suggesting that non-overlapping RNA populations may be affected; this would be consistent with data from human PAXT mutants, in which nuclear RNA accumulations have been shown to harbour RNAs that would usually be degraded by the exosome complex[104,105]. Interestingly, depletion of the exosome complex also causes read-through transcription and delayed transcription termination[137]. Metagene analysis of published RNAPII ChIP-seq data for *P.nmt-dis3* over the same set of genes reveals very similar changes in RNAPII occupancy as we have observed after Dbp2 depletion (Fig. 8C and Suppl. Fig. 8A)[137]. Moreover, in a comparative RNA interactome capture experiment in which we had previously compared occupancies of RNA-binding proteins on polyadenylated RNA between wild type and *mtl1-1* or *rrp6Δ* (mutants of MTREC and the exosome complex, respectively), CPAC components – in particular CFIA – were significantly enriched, suggesting that CPAC release might be impaired in exosome mutants[29,138] (Suppl. Fig. 8B, C). However, although the strong similarities between the phenotypes for *dbp2* and exosome mutants are suggestive, further research is necessary to develop a robust theory regarding their mechanistic connection.

### Comparative interaction profiling reveals sequence of co-transcriptional events

Although both Srp2 and Dbp2 are associated with chromatin across the entire transcription unit of RNAPII-transcribed genes, our data suggest that Srp2 associates with transcribing RNAPII earlier during the transcription cycle than Dbp2 and is also released earlier (Fig. 1C). Consequently, the relative enrichment scores of proteins in the comparative protein interaction profiling data of Srp2-HTP and Dbp2-HTP appear to reflect how early or late a factor is recruited to the RNAPII holocomplex during the transcription cycle. In general, subunits of the same complex tend to have very similar enrichment scores, significantly increasing our confidence in assigning scores to complexes. This proposed relationship between relative enrichment and time of arrival is consistent for many factors for which the timing of recruitment is known (Fig. 2C, D). It is therefore tempting to speculate that this dataset may also have predictive value for factors for which the order of recruitment is currently unclear. For example, the strong enrichment of TREX in the Srp2-HTP purification (Fig. 2D and Suppl. Fig. 2E) suggests that the mode of TREX recruitment in *S. pombe* may reflect the metazoan situation, where TREX is recruited to the 5'-end of genes in a splicing-dependent manner, in contrast to *S. cerevisiae*, where TREX is recruited by a transcription-coupled mechanism[139–141]. It is unclear to what extent the exon junction complex (EJC) in *S. pombe* resembles the mammalian complex and whether it can interact with

TREX in a manner similar to the human complex[142,143]; however, its relative enrichment score is similar to that of TREX, suggesting a possible interdependence in recruitment (Fig. 2D and Suppl. Fig. 2E). At the time, we are lacking high-resolution ChIP-seq data for these factors to validate these predictions. Nevertheless, we suggest that comparative interactome profiling of factors associated with different stages of transcription constitutes a valid experimental approach to determine the sequence of co-transcriptional events.

### ATP-dependence of RNA-protein complex dissolution as a general principle in RNA metabolism

Finally, we want to point out that in the field of splicing, the importance of DEAD- and DEAH-box ATPases in reversing the highly specific interactions involved in splice site recognition and stabilising the various conformations of the spliceosome – thereby enabling the highly dynamic, multi-step splicing reaction to proceed – has long been recognised[144]. Viewed in this context, it is not surprising that the release of CPAC might be coupled to ATP hydrolysis via the action of a DEAD-box ATPase, Dbp2. Whether the energy expenditure for complex dissolution serves as the cost for the high selectivity of the initial RNA-protein interaction or simply introduces an additional layer of regulation remains to be determined.

## Methods

### Yeast strains and manipulations

All *S. pombe* strains used in this study are listed in Supplementary Dataset 1. Standard methods were used for cell growth and genetic manipulations[145–150]. Oligos and plasmids used for strain construction are listed in Supplementary Dataset 2 and Supplementary Dataset 3, respectively. Sequences and functional annotations were retrieved from PomBase[43]. Cells were grown in yeast extract with supplements (YES) or Edinburgh minimal medium with glutamate (EMMG) at 30 °C unless indicated otherwise.

### Antibodies

Details on antibodies used in this study are listed in Supplementary Dataset 4.

### Chromatin immunoprecipitation

ChIP-seq was carried out according to published protocols[129], with minor modifications. Exponentially growing cells (200 ml) were crosslinked with 1% formaldehyde for 20 min at room temperature (RT). For calibrated ChIP-seq of RNAPII, *S. cerevisiae* cells were added to the culture in a 1:5 $OD_{600}$ ratio prior to crosslinking. 30 ml of a solution of 3 M glycine, 20 mM Tris were added to quench the reaction. Cells were pelleted and washed once with cold TBS and once with FA lysis buffer/0.1% SDS (50 mM Hepes-KOH pH 7.5, 150 mM NaCl, 1 mM EDTA, 1% Triton X-100, 0.1% sodium deoxycholate). After resuspension in 1.4 ml FA lysis buffer/0.5% SDS, cells were distributed to two screw-cap microtubes containing 300 µl glass beads and lysed in a FastPrep instrument (MP Biomedicals) at 3 x 45 s 6 m/s, with 2 min breaks on ice. Lysed cells were recovered with FA lysis buffer/0.1% SDS and ultra-centrifuged (59,100 x g, 20 min, 4 °C) in a 70 Ti rotor. The chromatin pellet was resuspended in FA lysis buffer / 0.1% SDS and sheared with a Bioruptor sonicator (Diagenode) at 15 s ON/45 s OFF for 80 min. HTP-tagged proteins were immunoprecipitated with IgG-coupled Dyna-beads (M-280 Tosylactivated, cat. No. 14204, Thermo Scientific), RNAPII with antibody against Rpb1 (8WG16, Millipore) coupled to 20 µl of protein-G dynabeads (Life Technologies). After washing and elution of bound material from the beads, proteins were digested by incubation with 0.6 mg pronase for 1 h at 42 °C, followed by decrosslinking at 65 °C overnight and DNA extraction. Sequencing libraries were constructed with the TruSeq ChIP Library Preparation Kit (Illumina) and sequenced on a NextSeq 2000 instrument (Srp2-HTP & Dbp2-HTP) or with the NEBNext Fast DNA Library Prep Set for Ion Torrent Kit

(E6270L, NEB) and sequenced on an Ion Torrent Proton (Life Technologies) (RNAPII-ChIP).

## Comparative protein interactome profiling

Cells were grown in 2 l YES to an $OD_{600}$ of 3.9 and crosslinked with 0.01% formaldehyde for 10 minutes at 30 °C before harvesting. The experiment was carried out in parallel with three biological replicates. Cell pellets were washed once with lysis buffer (20 mM HEPES pH 8.0, 100 mM KAc, 2 mM $MgCl_2$, 3 mM EDTA, 0.1% NP-40, 10% glycerol, 1 mM DTT, protease inhibitor (P8215, Sigma) and snap frozen in liquid nitrogen until use. Cells were lysed by grinding under liquid nitrogen for 30 min, and lysates pre-cleared at 3488 x g for 12 minutes at 4 °C. Lysates were cleared by centrifugation for 1 h at 207,800 x *g* at 4 °C (Optima XPN-80 Ultracentrifuge with 70 Ti, Beckman Coulter). HTP-tagged proteins were immunoprecipitated with 600 µl pre-washed IgG-sepharose 6 fast flow affinity resin (GE17-0969-01, Cytiva) for 2 h at 4 °C on a rotating wheel. After washing, proteins were eluted by TEV cleavage (20 µg TEV in 200 µl lysis buffer for 75 min at 16 °C) with 20 µg home-made TEV enzyme. The TEV eluate was incubated with 200 µl pre-washed Protino® Ni-NTA Agarose beads (745400.100, Macherey-Nagel) for 1 h at 4 °C on a rotating wheel. After washing (20 mM HEPES pH 8.0, 250 mM NaCl, 10 mM imidazole), purified proteins were eluted in elution buffer (20 mM HEPES pH 8.0, 150 mM NaCl, 300 mM imidazole) for 30 minutes at 4 °C on a rotating wheel.

## Detection of protein interactors by liquid chromatography-mass spectrometry

The eluate was then precipitated using 7x volumes acetone for at least 2 h at −20 °C. Following centrifugation, the protein pellet was washed twice with 300 µl ice-cold acetone. The protein pellet was then dried and reconstituted in 100 mM ammonium bicarbonate containing 1 mM TCEP and incubated at 90 °C for 10 min. Alkylation of reduced disulphide bonds was performed with 5 mM iodoacetamide for 30 min at 25 °C in the dark. Proteins were digested with 1 µg trypsin at 30 °C overnight. The sample was then acidified using trifluoroacetic acid and peptides were purified using Chromabond C18 Microspin columns (730004, Macherey-Nagel).

Dried peptides from on-bead digests were reconstituted in 0.1% trifluoroacetic acid and then analysed using liquid chromatography mass spectrometry carried out on a Q-Exactive Plus instrument connected to an Ultimate 3000 RSLC nano and a nanospray flex ion source (all Thermo Scientific). Peptide separation was performed on a reverse phase HPLC column (75 µm x 42 cm) packed in-house with C18 resin (2.4 µm; Dr. Maisch). The following separating gradient was used: 98% solvent A (0.15% formic acid) and 2% solvent B (99.85% acetonitrile, 0.15% formic acid) to 35% solvent B over 36 minutes at a flow rate of 300 nl / min. The data acquisition mode was set to obtain one high resolution MS scan at a resolution of 70,000 full width at half maximum (at $m/z$ 200) followed by MS/MS scans of the 10 most intense ions. To increase the efficiency of MS/MS attempts, the charged state screening modus was enabled to exclude unassigned and singly charged ions. The dynamic exclusion duration was set to 30 s. The ion accumulation time was set to 50 ms (MS) and 50 ms at 17,500 resolution (MS/MS). The automatic gain control (AGC) was set to $3 \times 10^6$ for MS survey scan and $1 \times 10^5$ for MS/MS scans.

Due to an instrument upgrade, co-IP peptide samples generated as above were analysed on an Exploris 480 (Thermo Scientific) with identical LC settings. The gradient was adjusted to separate peptides with 2% solvent B to 35% solvent B over 30 min. MS settings were set as follows: Spray voltage 2.3 kV, ion transfer tube temperature 275 °C, the mass of m/z 445.12003 was used as internal calibrant, MS1 resolution was set to 60,000 with max ion injection time of 25 ms and an automatic gain control (AGC) of 300%. The RF lens was 40%. MS/MS scans (cycle 1 s) were carried out with an Orbitrap resolution of 15,000 with

an ACG setting of 200%, quadrupole isolation was 1.5 m/z, collision was induced with an HCD collision energy of 27%.

MS raw data was then analysed with MaxQuant (2.0.3.0)[151], and a *S. pombe* uniprot database. MaxQuant was executed in standard settings (Supplementary Dataset 5) without "match between runs" option. The search criteria were set as follows: full tryptic specificity was required (cleavage after lysine or arginine residues); two missed cleavages were allowed; carbamidomethylation (C) was set as fixed modification; oxidation (M), deamidation (N,Q), dimethyl (KR), phospho (STY) as variable modifications.

The MaxQuant proteinGroup.txt file was further processed with SafeQuant (2.3.4)[152,153] to apply two-sided moderated t-statistics for p-value calculation[154] on MaxQuant LFQ values. Q-values were calculated according to the Benjamini-Hochberg method.

## Immunofluorescence

Immunofluorescence in Mtl1-GFP was carried out based on a published protocol[155]. 9 ml cell culture were fixed by the addition of 1 ml 37% formaldehyde for 30 min at RT on a rotating wheel. Cells were washed three times with PEM (100 mM piperazine-N,N′-bis(2-ethanesulfonic acid) (PIPES), 1 mM EGTA pH 8.0, 1 mM $MgSO_4$, pH 6.9) and spheroplasted in 500 µl PEMS (PEM with 1.2 M sorbitol) with 12.5 µl 10 mg/ml zymolyase T100 (9329.2, Carl Roth) for 1 h at 37 °C. Fixed cells were permeabilised in 500 µl PEMS, 1% Triton X-100 for 30 s, washed three times with PEM, then rotated for 30 min in 200 µl PEMBAL (PEM with 1% bovine serum albumin, 100 mM lysine hydrochloride, 0.1% $NaN_3$) at RT. 1/5th of the total fixed cell pellet was used per condition. Cells were incubated in 100 µl PEMBAL with anti-RNA polymerase II CTD repeat YSPTSPS (phospho S2) (Abcam, ab252855, 1:1000) at 4 °C overnight. Cells were washed twice with PEMBAL and incubated in 200 µl PEMBAL for 30 min at RT. Cells were incubated in 100 µl PEMBAL with Alexa Fluor 594-conjugated AffiniPure Goat Anti-Rat (Biozol, JIM-112-585-003, 1:400) at 4 °C overnight in the dark. Cells were washed once with PEMBAL and once with PBS, then resuspended in 40 µl PBS. 3 µl cell suspension were freshly mounted on poly-lysine-coated coverslips, let to dry, and imaged under ROTIMount FluorCare DAPI (HP20.1, Carl Roth).

## Fluorescence microscopy

Z-stack images were acquired on a Deltavision Ultra High-Resolution Microscope (Cytiva) using AcquireUltra (1.2.2) and deconvolved with softWoRx (7.2.1) using default settings. Line profiles were generated with Fiji (ImageJ 1.53t)[156] and plotted in R. For live-cell imaging, cells were grown overnight, harvested, and mounted on poly-lysine-coated coverslips in EMMG medium. Unless indicated otherwise, one individual z-plane is shown.

Image quantification was carried out with Fiji (ImageJ 1.53t) using a semiautomated ImageJ macro script (Supplementary Software 1). In short, cellular segmentation was carried out by thresholding on the transmitted light channel (cell outlines) and DAPI stain or NLS-BFP signal (nuclei). Measurements of oligo-$dT_{50}$ signal were performed on average intensity Z-projections for nucleus and cytoplasm (= total cell without nucleus) and the ratio of mean nuclear fluorescence intensity over mean cytoplasmic fluorescence intensity calculated for each cell. Quantitation of cleavage body proteins (Pcf11-GFP and Rna15-GFP) in live cells was performed on maximal intensity Z-projection and the maximal fluorescence intensity scored for each cell. The displayed p-values for the pair-wise comparisons were calculated using a two-sided Wilcoxon rank sum test and the ggpubr package (0.6.0) in R[157].

## Fluorescence in situ hybridisation

Cells were grown to an $OD_{600}$ of 0.5-0.7. For heat shock samples, 5.5 ml cell culture were mixed with 3.3 ml pre-warmed medium, then shifted to 42 °C for 1 h under agitation. Cells were fixed by the addition of 1.25 ml 37% formaldehyde at 42 °C for 20 min, then for an additional

1.5 h at RT on a roller mixer. For non-heat shock samples, 8.75 ml cell culture were fixed with 1.25 ml 37% formaldehyde for 1.5 h at RT. Cells were harvested (5 min, 3000 x $g$) and washed with 5 ml 0.1 M KPO$_4$, pH 6.4. Cells were then transferred to 1.5 ml tubes and incubated in 1 ml wash buffer (0.1 M KPO$_4$ with 1.2 M sorbitol) at 4 °C overnight. Cells were spheroplasted in 200 µl wash buffer containing 100 µg zymolyase T100 (9329.2, Carl Roth) for 1 h at 37 °C, then washed once with 1 ml wash buffer. Cells were incubated in 200 µl 2 x SSC at RT for 10 minutes, then resuspended in pre-hybridisation buffer (50% formamide, 10% dextran sulphate, 125 µg/ml $E.$ $coli$ tRNA, 500 µg/ml herring sperm DNA, 1 x Denhardt's solution) and incubated for 1 h at 37 °C. 1 µl of 1 pmol/µl Cy3-labelled oligo-d(T)$_{50}$ probe was added to the sample mix and incubated at 37 °C overnight in the dark. Cells were pelleted, gently resuspended in 600 µl of 0.5 x SSC and incubated on a rotating wheel for 30 minutes in the dark at RT. Cells were washed with 600 µl PBS, then resuspended in 50 µl PBS containing 0.1% NaN$_3$. ½ volume was mounted on a poly-lysine-coated coverslip and left to sit for 30 min at RT in the dark. Non-adherent cells were removed by aspiration and cells then imaged under ROTIMount FluorCare DAPI (HP20.1, Carl Roth) using a Deltavision Ultra High-Resolution Microscope (Cytiva).

## SDS lysis and extract preparation
100 ml cultures were grown to an OD$_{600}$ of 1.0 and harvested by centrifugation (2 min, 3000 x $g$ at 4°C). Cell were washed with 1 ml ice-cold H$_2$O and transferred into 2 ml microcentrifuge tubes before lysis with 400 µl glass beads (425-600 µm Ø) in 150 µl lysis buffer (10 mM Tris-HCl pH 8.0, 200 mM KCl, 2.5 mM MgCl$_2$, 0.5 mM EDTA pH 8.0, 0.5% NP-40, protease inhibitor cocktail (Sigma Merck, P8215), 1 mM PMSF) by vortexing for 20 min with 30 s on/off time intervals on ice. Lysates were transferred to a fresh microcentrifuge tube and cleared by centrifugation (5 min, 21,000 x $g$ at 4 °C). Protein concentration was determined by BCA Protein Assay (71285-M, Novagen) on a NanoDrop Microvolume Spectrophotometer (Fisher Scientific). 10 µg total protein per lane were resolved on a 10% SDS-PAGE gel.

## RNase/DNase treatment of crude lysates
Crude SDS lysates were prepared as above without the clearing spin. 50 µl aliquots containing 1 mg total protein were distributed into 1.5 ml microcentrifuge tubes and treated with either 5 µl RNase mix (100 µl RNase A (100 g/ml, Applichem), 100 µl RNase T1 (1000 U/µl, Thermo Scientific), 2 µl RNase III (1 U/µl, Invitrogen), 1 µl RNase H (5000 U/µl, NEB), 50 µl RNase I (100 U/µ, Jena Bioscience), 5 µl DNase I (1800U, Invitrogen), or incubated without enzymes for either 30 min at 4 °C or 15 min at 37 °C. Lysates were cleared by centrifugation (5 min, 21,000 x $g$ at 4 °C) and analysed by SDS-PAGE and Western blot.

## Co-immunoprecipitation assays
Lysates were prepared from 240 OD$_{600}$ units as described above. 1 mg total proteins were incubated with 15 µl pre-equilibrated magnetic ChromoTek GFP-TRAP beads (gta, proteintech) for 2 h on a rotator at 4 °C. Beads were washed 6 x with 1 ml lysis buffer for 5 min at 4 °C. Bound protein was then eluted with 60 µl HU loading buffer (8 M urea, 5% SDS, 200 mM Tris-HCl pH 6.5, 20 mM dithiothreitol (DTT), 1.5 mM bromophenol blue) for 5 min at 95 °C. 20 µl per co-IP and 10 µl of input (1:10) were separated on a 10% SDS-PAGE gel.

## TCA lysis and extract preparation
Trichloroacetic acid (TCA) lysis was carried out based on a published protocol[110]. 50 ml cultures were grown to an OD$_{600}$ of 0.5–0.6 and harvested by centrifugation (2 min, 3000 x $g$ at 4°C). Cells were washed with 1 ml ice-cold H$_2$O and transferred into 2 ml microcentrifuge tubes. Pellets were washed once with ice cold 20% TCA (15 s, 21,000 x $g$ at 4°C), and then lysed with 500 µl glass beads (425–600 µm Ø) in 500 µl 20% TCA by vortexing 3 x 1 min with 1 min breaks on ice.

The bottom of the tube was punctured with a hot 22-gauge ½ inch needle and lysates were transferred into new 1.5 ml microcentrifuge tubes by spinning (2 min, 376 x $g$ at 4°C). Beads were washed once with 600 µl ice-cold 5% TCA and the wash pooled with the lysates. Precipitated proteins were pelleted by centrifugation (10 min, 18,000 x $g$ at 4°C). Pellets were washed with 1 ml ice-cold 100% ethanol (5 min, 21,000 x $g$ at 4°C). Pellets were resuspended in 84 µl 1 M Tris pH 10.0 and 167 µl HU loading buffer (8 M Urea, 5% SDS, 200 mM Tris-HCl pH 6.5, 20 mM DTT, 1.5 mM bromophenol blue) and boiled for 5 min at 95°C before sample separation on a 10% SDS-PAGE gel (15 µl per lane).

## RNA preparation and Northern Blot
RNA was prepared with the hot phenol method and resolved on 1.2% agarose gels after glyoxylation according to published protocols[158]. For strand-specific Northern blot, digoxigenin (DIG)-labelled probes were generated with the DIG RNA labelling kit (11175025910, Roche) and detected using the DIG system (Roche). Oligos for probe preparation are listed in Suppl. Dataset 2. In all cases, methylene blue staining of ribosomal bands served as a loading control.

## RT-PCR amplification using an anchored (T)$_{12}$VN primer to map poly(A) site usage
Analysis of poly(A) site usage was carried out based on a published protocol[159]. We used the TVN-PAT primer to prime a reverse transcription reaction with SuperScript III Reverse Transcriptase (18080093, Invitrogen) on total RNA, followed by PCR amplification with a gene-specific forward primer and a reverse primer corresponding to sequences contained in TVN-PAT. Oligo sequences are listed in Suppl. Dataset 2.

## RNA sequencing
For RNA sequencing, pellets from $S.$ $pombe$ and $S.$ $cerevisiae$ cultures were pooled in a 5:1 OD$_{600}$ ratio immediately before cell lysis. Total RNA was extracted with the hot phenol method. Libraries were prepared with the VAHTS Stranded mRNA-seq Library Prep Kit (Vazyme Biotech) after poly(A) selection and sequenced on a NextSeq 2000 system (Illumina) or using the TruSeq protocol (Illumina) after ribodepletion with the Ribo-Zero Magnetic Kit for yeast (Epicentre) and sequenced on an Illumina HiSeq 4000 instrument.

## High-throughput sequencing data analysis
After read trimming with Trimmomatic (0.39+galaxy2), RNA-seq data were aligned to versions ASM294v2 and R64-1-1 of the $S.$ $pombe$ and $S.$ $cerevisiae$ genome using the RNA STAR aligner (2.7.11a+galaxy0) on the Galaxy server[160–162]. All reads aligning to both yeasts were filtered out using FilterSamReads (Galaxy version 3.1.1.0), and the total number of reads aligning to $S.$ $cerevisiae$ used for normalisation of the $S.$ $pombe$ data[163]. Differential gene expression analysis was carried out with DESeq2 (2.11.40.8+galaxy0) after read counting with htseq-count (2.0.5+galaxy0) in Union mode on the ASM294v2.57 gene model provided by EnsemblFungi[164–166]. ChIP-seq data were aligned with Bowtie2 (2.5.3+galaxy1)[167] and processed as above. Genomic ranges for metagene plots and heatmaps were selected using the packages rtracklayer (1.58.0) and GenomicAlignments (1.34.1) in R Studio (2023.12.1), and metagenes generated with the metagene2 package (1.14.0) without normalisation before geometric scaling[168–170]. GO term annotations were retrieved using biomaRt (2.54.1)[171]. Plots were generated with the ggplot2 package (3.5.1) in R[172].

## Interaction prediction
Interaction between Srp2 and Rpb8 was predicted with ColabFold (v.1.3.0)[173], which couples AlphaFold2 with Google Collaboratory, using model type Alphafold2-multimer-v2 with default parameters, and visualised using predicted aligned error[174,175].

## Reporting summary

Further information on research design is available in the Nature Portfolio Reporting Summary linked to this article.

## Data availability

Raw (fastq) and processed (bedgraph) sequencing data generated in this study can be downloaded from EBI ArrayExpress with the accession numbers, E-MTAB-13714 and E-MTAB-13717. The mass spectrometry proteomics data have been deposited to the ProteomeXchange Consortium via the PRIDE partner repository[176] with the dataset identifier PXD048560. The following published data have been included in our analyses for comparison: E-MTAB-2237[137], PRJEB7403[177], GSE73144[67], GSE148799[138], GSE174347[111] and PXD016741[29]. Source data are provided with this paper.

## Code availability

The ImageJ macro script used for quantitation of RNA FISH and cleavage body intensity measurement in mixed cell experiments is provided as Supplementary Software 1. Updated versions of the script will be available on our public GitHub repository Kilchertlab/Microscopy-analysis [https://github.com/Kilchertlab/Microscopy-analysis].

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

## Acknowledgements

We thank the National BioResource Project Japan, François Bachand, Damien Hermand, Tomoyasu Sugiyama, and Lidia Vasiljeva (LV) for strains and constructs. This work was funded by the Deutsche Forschungsgemeinschaft (DFG, German Research Foundation) via the Emmy Noether Programme (KI1657/2-1 to CK), the priority programme SPP1935 (KI1657/3-1 to CK), and the graduate training group GRK2355 – Projektnummer 325443116 to CK. We are grateful to LV for support and advice, to Johanna Seidler for her input on fluorescence image quantitation, and to Andreas Diepold, Vera Bettenworth and members of the lab for their critical comments on the manuscript.

## Author contributions

E.A. and C.K. conceived and planned experiments. E.A., S.S., J.B., B.K., and C.K. carried out experiments. T.G. performed proteomics analysis. J.W. generated resources. E.A., B.Z., and C.K. were responsible for data analysis and visualisation. C.K. and E.A. wrote the manuscript with support from S.S., J.B., J.W., B.Z., and T.G.

## Funding

## Competing interests

The authors declare no competing interests.
