## [Peer Review File · Nature Communications]

DEAD-box ATPase Dbp2 is the key enzyme in an mRNP assembly checkpoint at the 3'-end of genes and involved in the recycling of cleavage factorsREVIEWER COMMENTS

Reviewer #1 (Remarks to the Author):

Remarks to the Author:

Aydin et al have investigated a role for Dbp2 in mediating mRNA release following the termination of transcription in fission yeast. Starting with genome-wide mapping of Dbp2, they found that this RNA helicase map primarily to the 3' end of genes in the termination window, compared to Srp2 that mapped to gene bodies. The interactome of Dbp2 revealed its association with many factors implicated in RNA processing and 3' end formation, as well as MTREC that is involved in RNA surveillance. Ddp2 was found to co-localize in cleavage bodies with MTREC subunits, although the proteins do not appear to be overlapping. Loss of Dbp2 induced growth defects, which were not rescued with MTREC subunits, except *iss10*. Dbp2 mutants also accumulated polyA+ RNA in the nucleus in cleavage bodies, which was exacerbated by MTREC mutations. Nuclear retention of polyA+ RNAs was suggested to lead to sequestration of CPAC subunits, leading to defects in cleavage and polyadenylation. Loss of Dbp2 furthermore led to an accumulation of RNAPII in the termination window, as measured by ChIP-seq, interpreted as a delay in termination of transcription. Overall, the findings are of interest to the field. However, many of the claims made are quite speculative and require further substantiation.

In Fig 1, meta-analysis showing peak association of Dbp2 in the termination window is only shown for RPGs. The browser shots in fig 1a don't appear to fit this profile. Please show meta-analysis at protein coding genes (excluding RPGs).

Fig 2, are the interactions dependent on RNA? Is it possible to perform IP with endogenous proteins, without cross-linking? In the text, there is quite a bit of discussion about proteins that were not found in the protein complex (p12 to 13). The proteins identified by immunopurification are highly dependent on the conditions of the experiment. It's not particularly meaningful if certain factors were not identified under the given conditions. The text should be modified accordingly.

Fig 3, the nuclear localization of Dbp2 was suggested to be in 'cleavage bodies', because of a co-localization with MTREC subunits. As this is a very important point for the remainder of the study, it is important to better characterize the nuclear structures to which Dbp2 localizes. Indeed, the data suggests that MTREC and Dbp2 may occur in the same structures, but their localization is not especially overlapping, as would be expected for proteins in the same complex. How does this degree of co-localization fit with data from ChIP-seq and proteomics? Also, in general, the images shown do not have a very high resolution. As the authors are basing many conclusions on co-localization or otherwise, it would be better to have more resolute images. Similarly, the authors conclude this section of the text by saying that cleavage bodies likely represent storage compartments for 3' end processing factors. It's over reaching to make such a conclusion on a fairly limited series of microscopy experiments.

Fig 4, it appears in the top panel that *Iss10* mutant gives a growth advantage in wild type conditions. Is the effect then specific to Dbp2-depleted cells?

Fig 5 addresses the recycling of CPAC components. In wt cells, Pcf11 co-localized with Red1 in cleavage bodies although, at least based on the quality of the images shown, the 2 proteins seem to localize to distinct spots. Next, in the absence of Dbp2, Pcf11 was suggested to relocalize to the nucleoplasm. However, loss of Dbp2 significantly modifies the cleavage body structures. It's not clear

from the images what or where the cleavage body is under these conditions. The authors conclude from the experiments in Fig 5a and b that Pcf11 has been redistributed from the storage compartment to the sites of RNA biogenesis. As for the previous section, the conclusion in the text is very overstated. More data should be provided to back up such claims. In Fig 5c, the authors should perform extractions in the presence and absence of DNase if they wish to conclude that the proteins are retained on chromatin, and +/- RNase if they want to say the proteins are retained on RNA.

Fig 6, the authors should use specific probes to show the 3' extension. The results should be confirmed by RT-PCR across the cleavage site. Re-initiation after the 3' end cannot be excluded. What is the frequency of the 3' end defect for a given transcript? The browser shots also show a defect in splicing. How does this contribute to the defects in termination?

Fig 7, the slow transition of RNAPII would be expected to favor the use of proximal PAS, not distal ones as suggested (but not confirmed) by RNA-seq. While depletion of CPAC subunits might be an explanation for the unexpected result, it remains speculative. Additional data should be provided to back up the claim.

Reviewer #2 (Remarks to the Author):

This is an interesting and well-performed paper that should be published in Nature Communications. It demonstrates new functions of Dbp2 and a reasonable model in which Dbp2 activity has a key role in the RNP assembly checkpoint at 3' ends, specifically the connection between mRNA export and release of CPAC components from chromatin. The evidence that Dbp2 interacts with CPAC and export factors, localizes to cleavage bodies, is important to remove poly(A)-containing RNAs from chromatin is solid. The physical evidence that loss of Dbp2 results in depletion of CPAC components from the soluble pool is convincing, and this is strongly supported by the skipping of poly(A) sites and delayed termination, which would be predicted from lower levels of CPAC activity. I have only a few minor comments

The localization experiments with Rpb1-Ser2P are overinterpreted. While S2P levels are highest around the 3'UTR, they are still high over much of the coding regions. So, it is not a good marker for the 3'

The data in the FISH experiments is sometimes hard to visualize. Perhaps this is just a problem in copies instead of the ultimately published figures, but this should be checked.

The writing is quite long-winded in all sections of the paper. It could be condensed considerably by eliminating information/words that are irrelevant or unimportant for the main point of the paper. This wordiness makes it more difficult to understand the main findings. Also, readability would be significantly improved by subdividing the very long paragraphs into multiple ones. This would improve the logical thread.

Reviewer #3 (Remarks to the Author):

In eukaryotic cells, the expression of protein-coding genes depends on the production of functional mRNAs, which involves the formation of the 3'-end of the mRNA by endonucleolytic cleavage and

subsequent polyadenylation (CPA). The CPA complex (CPAC) carries out CPA. In this study, the authors report that Dbp2, an RNP remodeling ATPase belonging to the DEAD-box family, is the active component of an mRNP remodeling checkpoint that enables RNA export and is coupled to the release of CPAC. In addition to various cell biology and biochemical assays, the authors use comparative proteomics in an effort to identify the interaction of Dbp2 with CPAC and RNA export factors. This review is focused on the mass spectrometry-based proteomics aspects of the study.

In general, the study lacks sufficient description of the controls that were utilized for the proteomics-based protein-protein interaction studies, and there is a lack of detail regarding the data filtering criteria to account for false-positive identifications. Additional details regarding these major concerns are presented here:

1. P. 9, line 22: It would be informative to provide a description of the "stringent purification protocols" utilized by the authors that "did not result in the co-purification of significant amounts of proteins or protein complexes with Dbp2-HTP".
2. P. 12, line 23: A fold-enrichment of 1.03 of the RNA/DNA helicase Sen1 is mentioned. This is a surprisingly low level of "enrichment", which arguably should not be considered an enrichment at all.
3. Methods
 - a. There is no description of the controls that were utilized for the IP and cross-linking experiments. Non-specific binding is a significant confounder in most proteomics-based protein-protein interaction studies.
 - b. The numbers of biological and technical replicates for the proteomic experiments are not mentioned in the methods section, although this information does appear in the figure legends. It is strongly recommended to include this information in the methods section as well.
 - c. Details are lacking regarding the post-processing data analysis procedures that were applied to the MaxQuant protein identification results. What filtering criteria were applied? Was an FDR cutoff applied? Was the data normalized?
4. Consider moving Figure 2A to the Supplemental Material.
5. Figure 2D: Add lines indicating the fold-change and p-value based cutoffs. In the manuscript, it is unclear how the authors are determining "significance" in the context of "significantly enriched interacting proteins". Some proteins that are highlighted in this figure, namely Spt5 and Rbp1 appear to have rather low fold-changes and insignificant p-values.
6. Figure S2A: Add lines indicating the fold-change and p-value based cutoffs.
7. Figure S2B: Add titles to the bottom x-axis and the right y-axis.
8. Figure 2C: Consider altering the placement of the protein names to improve the interpretation of the data shown in this figure.

Reviewer #4 (Remarks to the Author):

I co-reviewed this manuscript with one of the reviewers who provided the listed reports. This is part of the Nature Communications initiative to facilitate training in peer review and to provide appropriate recognition for Early Career Researchers who co-review manuscripts

We thank the reviewers for their time and effort in reviewing this manuscript. Below, we provide a detailed, point-by-point response to the issues that were raised.

REVIEWER COMMENTS

Reviewer #1 (Remarks to the Author):

Remarks to the Author:

Aydin et al have investigated a role for Dbp2 in mediating mRNA release following the termination of transcription in fission yeast. Starting with genome-wide mapping of Dbp2, they found that this RNA helicase map primarily to the 3' end of genes in the termination window, compared to Srp2 that mapped to gene bodies. The interactome of Dbp2 revealed its association with many factors implicated in RNA processing and 3' end formation, as well as MTREC that is involved in RNA surveillance. Ddp2 was found to co-localize in cleavage bodies with MTREC subunits, although the proteins do not appear to be overlapping. Loss of Dbp2 induced growth defects, which were not rescued with MTREC subunits, except *iss10*. Dbp2 mutants also accumulated polyA+ RNA in the nucleus in cleavage bodies, which was exacerbated by MTREC mutations. Nuclear retention of polyA+ RNAs was suggested to lead to sequestration of CPAC subunits, leading to defects in cleavage and polyadenylation. Loss of Dbp2 furthermore led to an accumulation of RNAPII in the termination window, as measured by ChIP-seq, interpreted as a delay in termination of transcription. Overall, the findings are of interest to the field. However, many of the claims made are quite speculative and require further substantiation.

In Fig 1, meta-analysis showing peak association of Dbp2 in the termination window is only shown for RPGs. The browser shots in fig 1a don't appear to fit this profile. Please show meta-analysis at protein coding genes (excluding RPGs).

We would argue that you can see the 3'-bias of Dbp2 vs. Srp2 even in the browser shot if you take the direction of transcription into account (see overlay below; compare peaks marked with arrows). To make the browser shots easier to read, we removed positions of non-coding genes, which are usually expressed at very low levels.

We now also include metagene analysis for all protein-coding genes excluding RPGs in Supplemental Figure 1C. We did not filter on gene expression levels or gene length. In the complete gene set, presumably due to the more diverse nature of the genes, the resolution between initiating and elongating RNAPII is less sharp (compare S5P & S2P signal in the bottom panel). Nevertheless, a peak for Dbp2-HTP coinciding with the RNAPII peak in the termination window is also discernible for this gene set.

Fig. 1C

Fig. S1C

Fig 2, are the interactions dependent on RNA? Is it possible to perform IP with endogenous proteins, without cross-linking?

To address the reviewer's question, we performed additional Co-IP experiments under non-crosslinking conditions. Specifically, we included Co-IPs with the CFIA components Rna15 and Pcf11, which show a weak but reproducible interaction even in the absence of crosslinking (Figure S2F). To assess the RNA and/or DNA dependence of the interactions, we treated the lysates with RNase or DNase prior to pull-down. Although Dbp2 levels were not changed by the treatment, this significantly reduced the amount of Dbp2 we were able to recover in IPs (see example below; not included in the revised manuscript). At the moment, we are not able to make any conclusive statement about the nucleic acid dependence of these interactions.

In the text, there is quite a bit of discussion about proteins that were not found in the protein complex (p12 to 13). The proteins identified by immunoprecipitation are highly dependent on the conditions of the experiment. It's not particularly meaningful if certain factors were not identified under the given conditions. The text should be modified accordingly.

We removed the portions of the text that discussed proteins that were not detected in the purification.

Fig 3, the nuclear localization of Dbp2 was suggested to be in 'cleavage bodies', because of a co-localization with MTREC subunits. As this is a very important point for the remainder of the study, it is important to better characterize the nuclear structures to which Dbp2 localizes. Indeed, the data suggests that MTREC and Dbp2 may occur in the same structures, but their localization is not especially overlapping, as would be expected for proteins in the same complex. How does this degree of co-localization fit with data from ChIP-seq and proteomics? Also, in general, the images shown do not have a very high resolution. As the authors are basing many conclusions on co-localization or otherwise, it would be better to have more resolute images

Our apologies about the image resolution. In part, this may have been caused by file conversion issues. We have now cropped the microscopy images in Figures 3 and 5 more closely around the cells to enlarge the features; in addition, we also repeated colocalization experiments with Red1 with a filter set that is better suited for imaging tdTomato (542 nm / 587 nm) and replaced the panels in Figure 3A. This has significantly improved resolution.

We also include additional data characterising the localisation of various cleavage body components in wild-type cells in Figure 3A, namely Red1 and Mtl1, Dbp2, Rrp6, Pab2, Rna15 and Msi2. Colocalization of Red1 with Mtl1, Rrp6 and Pab2 had been shown in the publications in which MTREC bodies were first characterised in *S. pombe* (Sugiyama & Sugioka-Sugiyama, 2011; Egan et al., 2014). We include the published strains because today's microscopes resolve subnuclear structures in much greater detail and allow to differentiate between factors with a preferential nucleolar localization (Rrp6 and Dbp2) and factors that are predominantly nucleoplasmic (Pab2, Rna15, Msi2) (see below). Nevertheless, we fully agree with the conclusions made previously: Regardless of differences in the general localization pattern, many cleavage factors (= all that we looked at) localize to foci that also contain Red1 or Mtl1, which perfectly colocalize and which we use as markers for the cleavage body compartment because they are the only components that exclusively localize to foci. We do not know whether there is an essential nucleating factor for cleavage bodies, but if there is, it is not Red1: In *red1D* or *iss10D* (which leads to loss of Red1 from foci (Yamashita et al., 2013)), localization of Dbp2 to foci is unchanged. We now include this data in Suppl. Fig. 4F.

3A

Similarly, the authors conclude this section of the text by saying that cleavage bodies likely represent storage compartments for 3' end processing factors. It's overreaching to make such a conclusion on a fairly limited series of microscopy experiments.

We now include additional data showing that reduced transcription leads to an accumulation of Pcf11-GFP and Rna15-GFP in cleavage bodies. For this, we use an analog-sensitive mutant of Cdk9, which leads to promoter-proximal stalling of RNAPII and elongation defects within minutes after addition of the inhibitor (Guigen et al, 2007; Booth et al., 2018). Under these conditions, fluorescent signal of both proteins in foci is significantly increased in the analog-sensitive strain, but not the wild-type control (Figure 3D and E), supporting the idea that cleavage bodies store idle 3'-end processing factors. We amended the text as follows: "Under these conditions, fluorescent signal of Pcf11-GFP and Rna15-GFP in foci is significantly increased in the analog-sensitive strain, but not the wild-type control (Figure 3D and E), supporting the idea that cleavage bodies are storage sites and may help to buffer CPAC levels. However, we cannot exclude that CPAC components within cleavage bodies contribute to processing of mRNAs that have been prematurely released or carry out re-cleavage of polyadenylated transcripts (Malika et al., 2017)."

3D

E

Fig 4, it appears in the top panel that *Iss10* mutant gives a growth advantage in wild type conditions. Is the effect then specific to *Dbp2*-depleted cells?

We do indeed observe slightly larger colonies for the *iss10Δ* strain compared to the wild type. We amended the text accordingly: "combination with *iss10Δ* provided *P.nmt-dbp2* with a mild but reproducible growth advantage under depletion conditions; however, growth was also slightly improved in the wild-type background (Suppl. Figure 4C)."

Fig 5 addresses the recycling of CPAC components. In wt cells, *Pcf11* co-localized with *Red1* in cleavage bodies although, at least based on the quality of the images shown, the 2 proteins seem to localize to distinct spots. Next, in the absence of *Dbp2*, *Pcf11* was suggested to relocate to the nucleoplasm. However, loss of *Dbp2* significantly modifies the cleavage body structures. It's not clear from the images what or where the cleavage body is under these conditions. The authors conclude from the experiments in Fig 5a and b that *Pcf11* has been redistributed from the storage compartment to the sites of RNA biogenesis. As for the previous section, the conclusion in the text is very overstated. More data should be provided to back up such claims. In Fig 5c, the authors should perform extractions in the presence and absence of DNase if they wish to conclude that the proteins are retained on chromatin, and +/- RNase if they want to say the proteins are retained on RNA.

We now include quantitative data documenting the depletion of CFIA components *Pcf11* and *Rna15* from cleavage bodies in the absence of *Dbp2* (Figure 5A and B). In addition, we provide data supporting the redistribution of *Pcf11* from cleavage bodies to the RNAPII compartment marked by *Rpb9-iRFP* (Figure 5E and F). We have also carried out experiments where we treated the crude, pre-cleared lysates with RNase or DNase prior to removal of cellular debris by centrifugation. Treatment with DNase but not RNase led to a recovery of *Msi2-GFP* levels in the soluble fraction to levels comparable to those observed in the wild type. This data is now included in Figure 6C. The same experiment for *Pcf11-GFP* was inconclusive because incubation of the crude lysate led to recovery of soluble *Pcf11* even in the absence of enzymes (Supp. Figure 6A).

Fig 6, the authors should use specific probes to show the 3' extension. The results should be confirmed by RT-PCR across the cleavage site. Re-initiation after the 3' end cannot be excluded. What is the frequency of the 3' end defect for a given transcript? The browser shots also show a defect in splicing. How does this contribute to the defects in termination?

We now provide additional data that unambiguously shows that depletion of Dbp2 leads to alternative poly(A) site usage, with increased usage of distal sites and the production of 3' extended RNA species, including:

- RT-PCR amplification using an anchored (T)₁₂VN primer to map the sites of polyadenylation (Fig 7D).
- Northern blotting with a probe specific for the 3' extension (Fig 7C).

For *rpl2501*, there is indeed a low level of intron retention after Dbp2 depletion, which can be detected with a Northern blot probe against the intron (shown above; not included in the revised manuscript). *Rpl2501* transcripts with retained introns have a higher tendency to use the distal PAS (~70% use the proximal PAS, ~30% use the distal PAS). This agrees with previous publications that have demonstrated coupling between RNA processing events. Conversely, most 3'-extended *rpl2501* transcripts in Dbp2-depleted cells have no retained intron, with intron retention much less prevalent overall than proximal PAS skipping. In this, the Northern blot results are in perfect agreement with the RNA-seq data (Figure 7B). Moreover, we observe 3'-extended transcripts for many transcripts that do not have an intron (e.g. *asl1*, *tma19* and *rpl1001* shown in Figure 7A and Suppl. Figure 6D and E), making it extremely unlikely that PAS skipping is merely a consequence of intron retention.

Fig 7, the slow transition of RNAPII would be expected to favor the use of proximal PAS, not distal ones as suggested (but not confirmed) by RNA-seq. While depletion of CPAC subunits might be an explanation for the unexpected result, it remains speculative. Additional data should be provided to back up the claim.

As stated above, we have now analysed poly(A) site usage after Dbp2 depletion for one representative transcript, which confirmed that proximal sites are skipped in favour of more distal ones, leading to 3' UTR lengthening (Figure 7C and D). In our view, there is consensus in the literature that 3'UTR lengthening can be a consequence of depletion of core poly(A) factors. We now reference a recent review article that also makes this claim (Tang and Zhou, 2022).

To specifically confirm that this is also the case in *S. pombe*, we analysed PAS usage on *rpl2501* mRNA in an auxin-induced degron strain of Rna15 (Rna15-3xsAID) that is unable to grow in the presence of the auxin-analog 5ada-IAA (see below; not included in the revised manuscript). In agreement with data from other organisms, we observed increased usage of distal sites after incubation of the degron strain with 5ada-IAA. However, we decided not to include this data in the revised manuscript because the strain we had generated displayed 3'-end processing defects in the presence of the OstTIR construct even in the absence of the drug.

Reviewer #2 (Remarks to the Author):

This is an interesting and well-performed paper that should be published in Nature Communications. It demonstrates new functions of Dbp2 and a reasonable model in which Dbp2 activity has a key role in the RNP assembly checkpoint at 3' ends, specifically the connection between mRNA export and release of CPAC components from chromatin. The evidence that Dbp2 interacts with CPAC and export factors, localizes to cleavage bodies, is important to remove poly(A)-containing RNAs from chromatin is solid. The physical evidence that loss of Dbp2 results in depletion of CPAC components from the soluble pool is convincing, and this is strongly supported by the skipping of poly(A) sites and delayed termination, which would be predicted from lower levels of CPAC activity. I have only a few minor comments.

The localization experiments with Rpb1-Ser2P are overinterpreted. While S2P levels are highest around the 3'UTR, they are still high over much of the coding regions. So, it is not a good marker for the 3' end.

We never meant to imply that we have the resolution to distinguish between transcribing and terminating RNAPII in IF. Our apologies if this was misleading. We have changed the text to "We carried out immunofluorescence microscopy with an antibody against S2P-modified Rpb1 to detect transcriptionally active RNAPII in a strain where the MTREC component Mtl1 was C-terminally tagged with GFP".

The data in the FISH experiments is sometimes hard to visualize. Perhaps this is just a problem in copies instead of the ultimately published figures, but this should be checked.

This may have been a file conversion issue. We have double-checked that the original figures have a sufficient resolution and will seek to submit them in a suitable format.

The writing is quite long-winded in all sections of the paper. It could be condensed considerably by eliminating information/words that are irrelevant or unimportant for the main point of the paper. This wordiness makes it more difficult to understand the main findings. Also, readability would be significantly improved by subdividing the very long paragraphs into multiple ones. This would improve the logical thread.

We have edited the manuscript to remove excess information. Negative data concerning MTREC was moved to the supplements.

Reviewer #3 (Remarks to the Author):

In eukaryotic cells, the expression of protein-coding genes depends on the production of functional mRNAs, which involves the formation of the 3'-end of the mRNA by endonucleolytic cleavage and subsequent

polyadenylation (CPA). The CPA complex (CPAC) carries out CPA. In this study, the authors report that Dbp2, an RNP remodeling ATPase belonging to the DEAD-box family, is the active component of an mRNP remodeling checkpoint that enables RNA export and is coupled to the release of CPAC. In addition to various cell biology and biochemical assays, the authors use comparative proteomics in an effort to identify the interaction of Dbp2 with CPAC and RNA export factors. This review is focused on the mass spectrometry-based proteomics aspects of the study.

In general, the study lacks sufficient description of the controls that were utilized for the proteomics-based protein-protein interaction studies, and there is a lack of detail regarding the data filtering criteria to account for false-positive identifications. Additional details regarding these major concerns are presented here:
1. P. 9, line 22: It would be informative to provide a description of the "stringent purification protocols" utilized by the authors that "did not result in the co-purification of significant amounts of proteins or protein complexes with Dbp2-HTP".

Here, "stringent" was meant to signify "without crosslinking". Otherwise, the conditions were the same. We changed the text to clarify this: "Initial purification attempts without crosslinking did not result in the co-purification of significant amounts of proteins or protein complexes with Dbp2-HTP."

2. P. 12, line 23: A fold-enrichment of 1.03 of the RNA/DNA helicase Sen1 is mentioned. This is a surprisingly low level of "enrichment", which arguably should not be considered an enrichment at all.

The wording here was unfortunate. It should have said: "Sen1 copurified with Srp2 and Dbp2, with no significant enrichment in either purification compared to the other.". Anyhow, the sentence is not included in the revised manuscript because we have removed the entire section.

3. Methods

a. There is no description of the controls that were utilized for the IP and cross-linking experiments. Non-specific binding is a significant confounder in most proteomics-based protein-protein interaction studies.

There are relatively few antibodies available against *S. pombe* proteins, so we generally rely on tagged strains for co-IP experiments. As a general rule, we use a strain in which only the prey protein is tagged as the negative control (Suppl. Figure 2F). We now include this information in the figure legend. If the prey protein sticks to the beads in the absence of tagged bait protein, we optimize washing conditions to get rid of the background binding. In cases where this is not possible, we disregard the experiment. For the interaction profiling of Dbp2, we chose a protein that is known to reside in the same compartment (Srp2) as a reference rather than an untagged strain. In our experience, untagged controls are not as informative because they tend to contain the same set of highly abundant cytoplasmic proteins.

b. The numbers of biological and technical replicates for the proteomic experiments are not mentioned in the methods section, although this information does appear in the figure legends. It is strongly recommended to include this information in the methods section as well.

We have added this information to the methods section.

c. Details are lacking regarding the post-processing data analysis procedures that were applied to the MaxQuant protein identification results. What filtering criteria were applied? Was an FDR cutoff applied? Was the data normalized?

We now provide the full parameter list used to analyse the raw data with MaxQuant in Suppl Table 5. No further normalization was applied to the results. We have added this information to the methods section.

4. Consider moving Figure 2A to the Supplemental Material.

Was done (now S2A).

5. Figure 2D: Add lines indicating the fold-change and p-value based cutoffs. In the manuscript, it is unclear how the authors are determining "significance" in the context of "significantly enriched interacting proteins". Some proteins that are highlighted in this figure, namely Spt5 and Rbp1 appear to have rather low fold-changes and insignificant p-values.

The fact that Spt5 and Rbp1 purify equally well with both Srp2 and Dbp2 (while the kinases that modify them at different stages of transcription do not) is exactly what this figure is meant to convey. We now added Spt5 to Figure 2A, which directly compares protein intensities from both purifications, and referenced the plot showing absolute protein intensities in the text to make this clear.

6. Figure S2A: Add lines indicating the fold-change and p-value based cutoffs.

Were added (now S2B), also to Figures S2D-F.

7. Figure S2B: Add titles to the bottom x-axis and the right y-axis.

Were added (now S2C).

8. Figure 2C: Consider altering the placement of the protein names to improve the interpretation of the data shown in this figure.

We have revised the figure to enhance clarity (now 2B).

Reviewer #4 (Remarks to the Author):

Thank you for taking the time to review the manuscript.

REVIEWERS' COMMENTS

Reviewer #1 (Remarks to the Author):

The authors have added a significant amount of data that has addressed the concerns raised in the initial review.

Reviewer #2 (Remarks to the Author):

In my initial review, I said that this is an interesting and well-performed paper that should be published in Nature Communications with minor modifications. The authors have satisfactorily addressed my comments. In my opinion, they have also satisfactorily addressed the more technical comments of the other reviewers. As such, I recommend that the paper be accepted for publication in Nature Communications.

Reviewer #3 (Remarks to the Author):

Some of the text provided by the authors in response to comment 3a from Reviewer #3 would be helpful to include in the Results section given the general concerns regarding appropriately controlling for non-specific protein binding in IP-mass spec experiments. Otherwise, the authors have provided comprehensive responses to the reviewers' comments, and the quality of the manuscript has been improved accordingly.

Reviewer #4 (Remarks to the Author):

We thank the reviewers for their time and effort in reviewing this manuscript. Below, we provide a point-by-point response to the issues that were raised.

REVIEWERS' COMMENTS

Reviewer #1 (Remarks to the Author):

The authors have added a significant amount of data that has addressed the concerns raised in the initial review.

Reviewer #2 (Remarks to the Author):

In my initial review, I said that this is an interesting and well-performed paper that should be published in Nature Communications with minor modifications. The authors have satisfactorily addressed my comments. In my opinion, they have also satisfactorily addressed the more technical comments of the other reviewers. As such, I recommend that the paper be accepted for publication in Nature Communications.

Reviewer #3 (Remarks to the Author):

Some of the text provided by the authors in response to comment 3a from Reviewer #3 would be helpful to include in the Results section given the general concerns regarding appropriately controlling for non-specific protein binding in IP-mass spec experiments. Otherwise, the authors have provided comprehensive responses to the reviewers' comments, and the quality of the manuscript has been improved accordingly.

As suggested by the reviewer, we have now included the following text in the results section of the manuscript:

"We chose a protein that is known to reside in the same compartment (Srp2) as a reference rather than an untagged strain (Suppl. Figure 2A). In our experience, untagged controls are not as informative because they tend to contain the same set of highly abundant cytoplasmic protein."

Reviewer #4 (Remarks to the Author):
